# Age structure landscapes emerge from the equilibrium between aging and rejuvenation in bacterial populations

Audrey M. Proenca [1,2], Camilla Ulla Rang[1], Christen Buetz[1], Chao Shi[1] & Lin Chao[1]

The physiological asymmetry between daughters of a mother bacterium is produced by the inheritance of either old poles, carrying non-genetic damage, or newly synthesized poles. However, as bacteria display long-term growth stability leading to physiological immortality, there is controversy on whether asymmetry corresponds to aging. Here we show that deterministic age structure landscapes emerge from physiologically immortal bacterial lineages. Through single-cell microscopy and microfluidic techniques, we demonstrate that aging and rejuvenating bacterial lineages reach two distinct states of growth equilibria. These equilibria display stabilizing properties, which we quantified according to the compensatory trajectories of continuous lineages throughout generations. Finally, we show that the physiological asymmetry between aging and rejuvenating lineages produces complex age structure landscapes, resulting in a deterministic phenotypic heterogeneity that is neither an artifact of starvation nor a product of extrinsic damage. These findings indicate that physiological immortality and cellular aging can both be manifested in single celled organisms.

[1] Section of Ecology, Behavior and Evolution, Division of Biological Sciences, University of California, San Diego, La Jolla, CA 92093, USA. [2] CAPES Foundation, Ministry of Education of Brazil, Brasilia, DF 70.040-020, Brazil. Correspondence and requests for materials should be addressed to A.M.P. (email: aproenca@ucsd.edu) or to L.C. (email: lchao@ucsd.edu)

Aging, defined broadly as the decline of function and consequent loss of fitness with time, is a ubiquitous characteristic of biological organisms[1,2]. Although bacteria were traditionally thought not to age, recent studies have suggested that the phenotype is present in asymmetrically dividing *Caulobacter crescentus* and *Escherichia coli*[3–6]. As rod-shaped bacteria such as *E. coli* divide at the middle, a new pole is synthesized at the division plane with every replication, while the distal old poles are conserved from the mother (Fig. 1a). Thus, each *E. coli* cell has an old and a new pole. Upon division, cells inheriting the maternal old pole are called old daughters, while the ones inheriting the newly synthesized pole are called new daughters. Old poles are consecutively inherited over generations, carrying accumulated non-genetic damage in the form of inclusion bodies, which are aggregates of misfolded proteins[6–9]. Old daughters, inheriting larger damage loads along with old poles, display a decline in growth rates associated with aging, while new daughters rejuvenate by receiving less damage.

Nonetheless, despite a succession of reports on bacterial aging[3–6,8–11] and the identification of similar asymmetry in other systems[12–15], the validity and significance of the phenomenon remains controversial. Although protein aggregates are strongly biased toward old poles in *E. coli* and reportedly correlate with functional decline[6,8,9,16], this association was often found to be equivocal in similar systems, such as fission yeast[13,17]. Studies in both *E. coli* and fission yeast suggested that aging is a consequence of extrinsic damage[11,13], configuring their divisional asymmetry as a conditional strategy[18] as opposed to a deterministic process. Moreover, studies following up on the first reports of bacterial aging found no reduction in the growth rate of cell lineages over hundreds of generations and no physiological asymmetry between old and new daughters[19]. Finally, even when deterioration or growth rate decreases were observed, the decline could be accounted by starvation. The problem arises because agar pads[4,6,11] and small microfluidic devices (such as the mother machine[19,20]) were used to sustain the bacteria during time-lapse microscopy. With agar pads, bacteria form mini-colonies and cells located in the center could become nutrient-limited within a few generations. With the mother machine, old daughters always reside at the blind end of growth channels, which is most removed from the nutrient source at the opposite and open end. Therefore, starvation could be present in previous bacterial aging studies.

Despite these conflicting results on bacterial aging, diverse studies have suggested asymmetric damage partitioning as a common mechanism of cell maintenance. Besides bacteria and

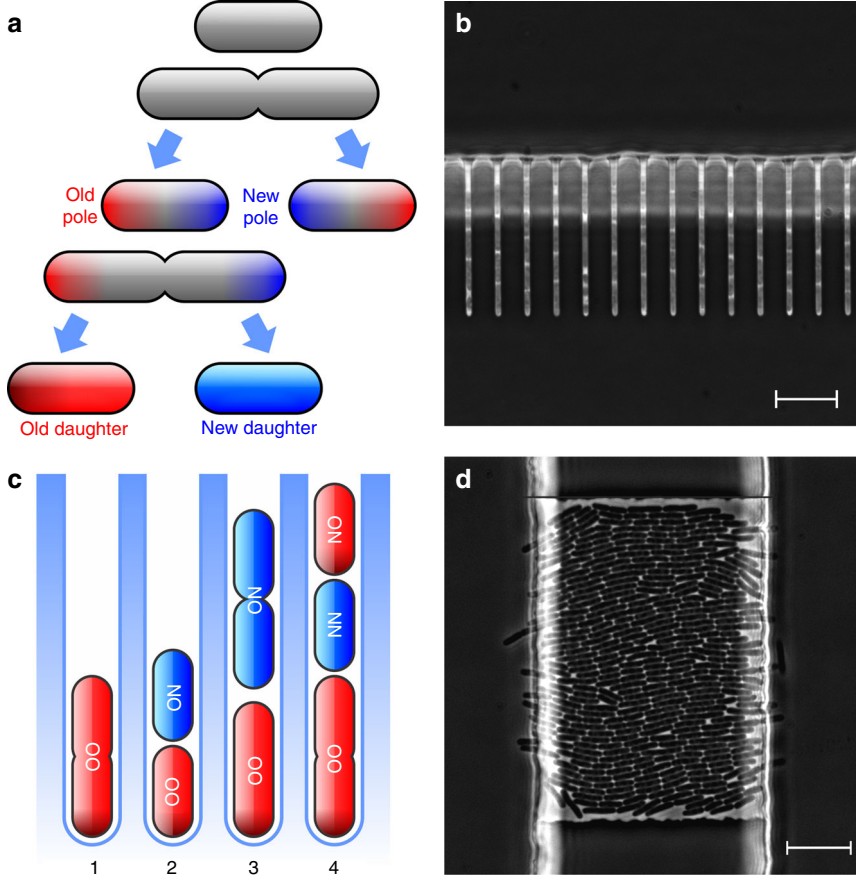

**Fig. 1** Polarity structure of rod-shaped cells in microfluidic devices. **a** Upon division, *E. coli* inherit a conserved old pole, along with a newly synthesized pole formed at the site of fission. On the next division, the old pole is again segregated to one sibling, which is called an old daughter, while the maternal new pole is inherited by the other sibling, called a new daughter. Old poles are consecutively inherited throughout generations, carrying accumulated non-genetic damage. **b** The mother machine design allowed imaging of ~30 growth wells per experiment, with each well harboring an old daughter lineage. Large flow channels (top) provided fresh nutrients to the traps, with the flow preventing the formation of biofilms on the device. **c** Within the mother machine traps, the structure of bacterial lineages maintains a constant pattern. The oldest cells (OO) remain at the closed end, generating another daughter like itself and a new daughter (ON) upon division. When this new daughter divides, its old daughter (NO) will be located by the opening of the well, therefore closer to the nutrient source than its young sibling (NN). A doubling time asymmetry generated solely by starvation would predict that NO cells grow faster than NN. **d** The daughter device consisted of large growth chambers, flanked by two wide flow channels providing fresh medium to the colony. As two-dimensional colonies can grow feely in this device, the lineages exhibit no rigid polarity structure. Scale bars = 10 μm

yeast, the physiological asymmetry between daughter cells has been observed in diatoms[12], nematodes[21], stem cells[22–24], and others. Thus, the identification of a deterministic divisional asymmetry in bacteria, leading to aging and rejuvenation in clonal populations, could further characterize aging at the cellular level as a ubiquitous process in living organisms.

In this study, we show that deterministic age structures emerge within single populations of unstressed *E. coli*, while maintaining long-term growth stability and proliferative immortality. We employed two microfluidic designs to ensure culturing of bacteria in the absence of extrinsic damage and to avoid starvation. Our results indicate that the asymmetry between new and old daughters does not correspond to differential nutrient deprivation. By following bacterial populations over several generations, we show that new and old lineages stabilize around two distinct growth equilibrium attractors, thus exhibiting stable growth over time while displaying consistent asymmetry. Moreover, sister lineages are constantly generated from these equilibria. With every division, a new mother in equilibrium generates a new daughter similar to itself, and an old daughter that loses function over generations as it ages toward the equilibrium attractor of old lineages. The opposite pattern is verified when new daughters generated from old mothers rejuvenate towards their own equilibrium attractor. Therefore, constant patterns of aging and rejuvenation connect distinct growth equilibria within clonal populations, providing evidence for deterministic age structures in bacteria. These results suggest that key aspects of biological aging may have originated in single-cell organisms, such as bacteria. We propose that the emergence of age structures allowed bacteria to evolve a more complex life history by adapting different stages to different ecological challenges.

## Results

**Physiological asymmetry does not derive from starvation**. To determine the presence of asymmetric damage partitioning in the absence of extrinsic damage, and to test whether starvation—rather than aging—could account for the difference between old and new daughters, we measured bacterial growth in microfluidic devices. Our measurements are hereby presented as doubling times, corresponding to elongation rates, which represents a physiologically meaningful parameter, while exhibiting lower variance than division intervals (Supplementary Fig. 1). Our first microfluidic design, known as the mother machine[19,20] (Fig. 1b), is designed to trap cells in a narrow linear channel (oriented vertically for reference) with a blind end at the bottom. A constant supply of nutrient media flows horizontally, carrying away cells growing out of the traps and delivering nutrients through the open end of the features. Thus, the bottom cells are located farthest from the nutrient source and a starvation gradient could exist. An attractive feature of the mother machine is that the linear channel reduces the tracking of cells to a one-dimensional problem.

In our design of the mother machine, the growth traps harbored four cells for complete division cycles. These bacterial cells were ordered as old, new, new, and old daughters, and denoted as OO, ON, NN, and NO (Fig. 1c), as the channel is too narrow for the cells to switch places. OO represents the cell that was an old daughter the last two generations, ON is a new daughter born of an old mother, and so forth. We found that ON cells displayed faster doubling times than OO (two-tailed paired *t*-test, $t = -23.152$, df = 555, $p < 0.001$; Supplementary Fig. 2), reflecting the expected asymmetry between new and old daughters. However, the fact that ON is also closer to the nutrient source than OO (Fig. 1c) suggests that either starvation or asymmetry could produce this pattern. Adding the third cell

NN is still confounding because both starvation and aging predict that it has a shorter doubling time. It is only the inclusion of NO, and its comparison to NN, that distinguishes between the two explanations. Although aging predicts that the rank of doubling times is NN < NO, as NN is the new daughter, starvation predicts the opposite due to NO being closest to the nutrient source. Our measurements of these cells provided clear support for aging over starvation, with a pair-wise comparison finding NO to display a significantly longer doubling time than NN (Fig. 2a, Supplementary Fig. 2 and Supplementary Table 1).

The above result alone should eliminate starvation as the explanation for the observed physiological differences between old and new daughters, but we tested this conclusion even more rigorously. We employed a second microfluidic design[25,26], comprising large growth chambers and controlling for positional nutrient deprivation by opening both ends of the traps (Fig. 1d). In these chambers, the larger width spread cells into a monolayer of approximately 300 bacteria, which pushed against each other and shuffled positions as they elongated. Unlike the mother machine, this device did not preferentially retain either the old or the new daughter. Thus, we named this design, for the purpose of this study, the daughter device, and used it to test whether the doubling time relationship observed for OO, ON, NN, and NO in the mother machine could be replicated. To assemble a data set equivalent to Fig. 2a, we identified OO cells in the daughter device as those that had been old for at least two generations, and then followed them forward in time to obtain the series OO, ON, NN, and NO. The relationships measured for OO, ON, NN, and NO in the mother machine (Fig. 2a) and the daughter device (Fig. 2b, Supplementary Fig. 2 and Supplementary Table 1) were nearly identical. Most importantly, NO ($20.18 \pm 0.97$ min, mean ± SD, $n = 620$) still had a significantly longer doubling time than NN ($19.60 \pm 0.94$ min, $n = 620$) in the latter device. To ensure that the shuffling was effective, we measured cell positions in the daughter device and found that OO, ON, NN, and NO experienced the same average distances from the open sides and nutrient sources (Fig. 2c, d and Supplementary Data 1). Thus, these results indicate that bacterial populations display asymmetric physiology in the absence of both extrinsic damage and starvation.

The strongest evidence that aging rather than starvation accounts for the difference between old and new daughters comes from a comparison of the rank order of OO, ON, NN, and NO doubling times. In both microfluidic devices, the rank was OO > NO > ON > NN (Fig. 2a, b), indicating that bacterial aging is more quantitative than just new daughters growing faster than old daughters. Rather, as OO and NO are not equivalent old daughters (mother machine: $t = -6.831$, df = 119.72, $p < 0.001$; daughter device: $t = -7.876$, df = 1125.5, $p < 0.001$; one-tailed *t*-tests), history and time also matter. In the next sections, we explore and quantify the effects of time on bacterial aging.

**Determinism and stochasticity explain doubling time variance**. The relationship between the doubling times of mother and daughter cells can be examined by a phase plane on which the doubling times of the new (T1) and old (T2) daughters are plotted against the doubling time T0 of their mother (Fig. 3a and Supplementary Table 2). Due to the large variance present in the data, we observed minimal improvement of fit for different nonlinear models (Supplementary Fig. 3A). Mother machine data, thus analyzed through linear regression, revealed a positive relationship between T0 and the pooled doubling times of the daughters (Fig. 3a, black line; $\beta = 0.28$, $t = 13.34$, $p < 0.001$). Nonetheless, as mentioned above and in Supplementary Fig. 2, new and old daughters exhibited distinct doubling times. In the

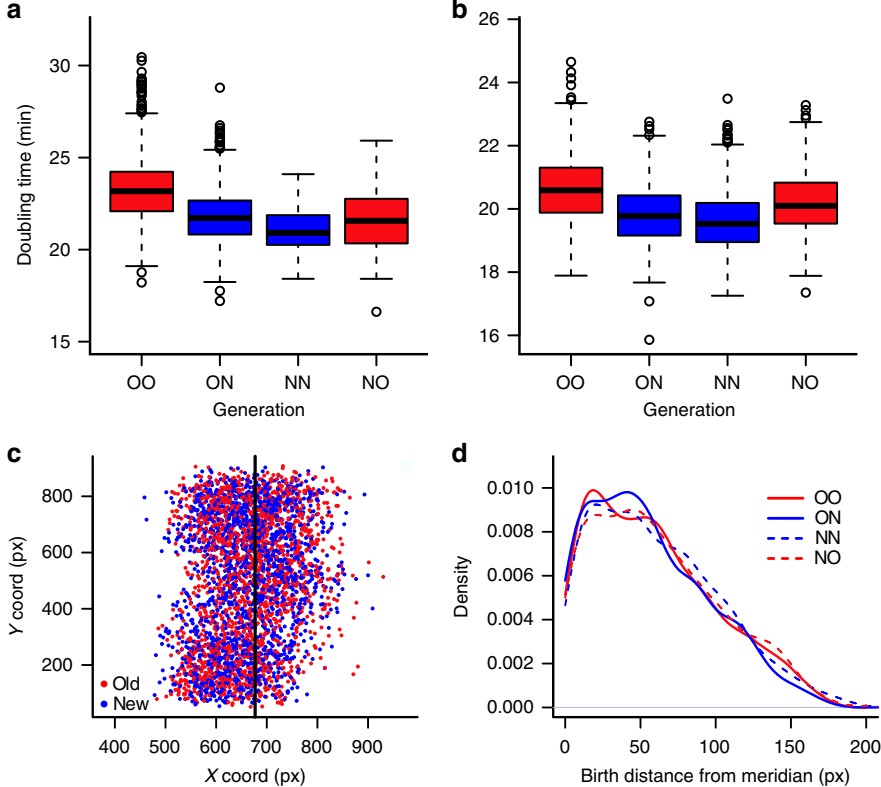

**Fig. 2** Doubling time asymmetry in the absence of starvation and stress. **a** Bacteria from the mother machine were categorized as OO, ON, NN, and NO (n = 882, 882, 105, 105 cells), reflecting their polarity and position within the growth traps. ON cells displayed faster doubling times than OO. Similarly, NN grew faster than NO (two-tailed paired t-test, t = − 2.308, df = 104, p = 0.023) despite the latter being closer to the nutrient source. **b** Doubling times of OO, ON, NN, and NO cells (n = 556, 556, 620, 620 cells) from the daughter device showed the same pattern as in the mother machine. The doubling time relationships OO > ON (t = − 17.219, df = 555, p < 0.001) and NO > NN (t = − 13.564, df = 619, p < 0.001) were again verified by two-tailed paired t tests. Boxplots show median (center line), first and third quartiles (box limits), and minimum and maximum (whiskers). **c** Representation of cells within the chamber, according to their coordinates at birth. The vertical line represents the midline of the chamber, which is open along the Y axis on both sides. **d** We analyzed the data from **c** by measuring the distance of each cell from the midline of the chamber at the moment of birth, and verified no localization bias among sibling pairs within the daughter device (OO–ON p = 0.12 and NO–NN p = 0.98, Wilcoxon signed-ranks test). Therefore, doubling time differences represent and effect of aging as opposed to starvation

phase plane, this physiological distinction produces a visual separation between new and old subpopulations.

To verify whether T1 and T2 subpopulations would be better explained by individual models, we performed a two-way analysis of covariance (ANCOVA) evaluating the effect of T0 and age (new or old) on daughter doubling times. Although both T0 (F = 209.15, p < 0.001) and age (F = 336.69, p < 0.001) had a significant effect on T1 and T2, there was also interaction between factors (F = 10.67, p = 0.001). This indicates that the relationship between T1 or T2 and T0 is best described by distinct slopes for each subpopulation. The independent models for new (linear regression; β = 0.22, t = 8.85, p < 0.001) and old daughters (β = 0.35, t = 11.44, p < 0.001) are shown in Fig. 3a. Thus, slow growing mothers produced daughters that were also slow, a pattern consistent with the prevalence of aging generated by asymmetric damage partitioning. Most importantly, the T2 regression line lay above and had a larger slope than the line for T1, which are expected signatures of aging affecting old daughters more.

Despite following the trends described by linear regression, new and old subpopulations displayed large variance. To determine the sources of this variance, we partitioned the sums of squared deviations for T1 and T2 doubling times (see Methods). We started by identifying the total sum of squares as the deviation of each doubling time from the population mean, obtaining $SS_T = 6737.66$. Some of this total variance was

produced by the positive relationship between a mother and its daughters ($\beta = 0.28$; Fig. 3b, bottom panels). This fraction of the variance, $SS_M$, was determined as the deviation of predicted pooled T1 and T2 from the population mean doubling time, 22.45 min. Thus, the sum of squared deviations due to maternal inheritance was obtained as $SS_M = 557.76$. Due to asymmetry, however, doubling times predicted by two separate linear models deviated from the values predicted by the mother alone (Fig. 3b, middle panels). This deviation, produced by asymmetry, was determined as $SS_A = 926.32$. Finally, observed doubling times deviated from predicted values due to stochasticity, with $SS_S = 5253.58$ (Fig. 3b, top panels). By combining these values, the fraction of the variance determined by deterministic sources is given as $(SS_M + SS_A)/SS_T = 0.22$, whereas the remaining variance was explained by stochastic factors $SS_S/SS_T = 0.78$. Daughter device populations obtained similar results, with 24.9% of the variation explained by non-genetic maternal inheritance and asymmetry (Supplementary Table 3).

These analyses suggest that the inheritance of damage across generations produces variance in doubling times of new and old daughters. Mother bacteria with longer doubling times likely carry larger accumulated damage loads, thus producing daughters with longer doubling times as well. Since these mothers partition damage asymmetrically upon division, asymmetry is also a source of deterministic variance in bacterial populations. In the following

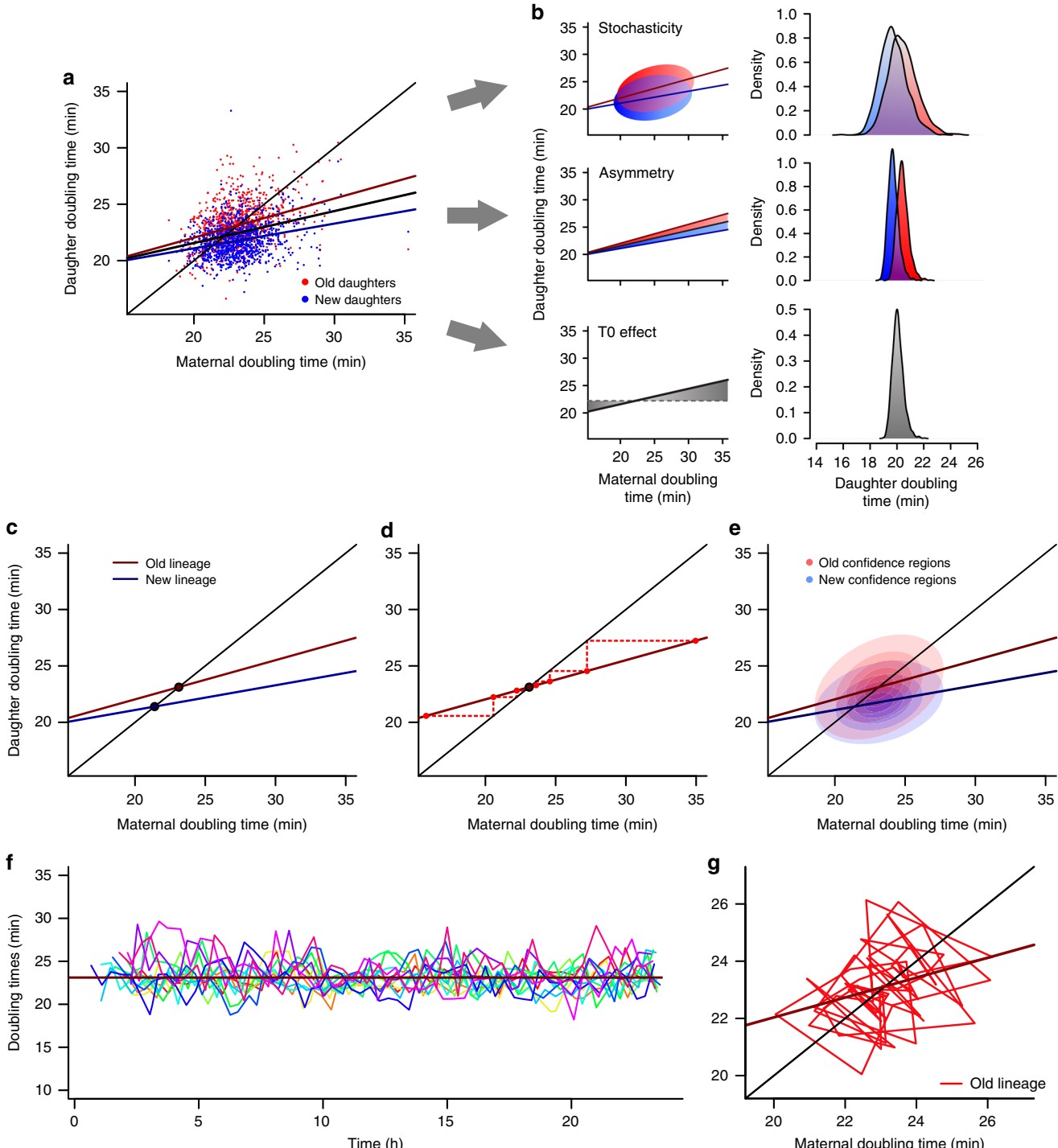

**Fig. 3** Doubling time relationships predict stabilization at equilibrium points. **a** Old daughters (23.127 ± 1.903 min, $n = 987$ cells) displayed longer doubling times than their new siblings (21.778 ± 1.517 min, $n = 987$ cells) in the mother machine ($t = -21.884$, df $= 986$, $p < 0.001$, paired one-tailed $t$-test), resulting in a separation between new and old subpopulations in the phase plane. **b** The variance in T1 and T2 was partitioned into three components through the estimation of sum of squared deviations. Left: shaded areas represent the variability explained by each component. Maternal doubling times explain the deviation from the population mean; asymmetry explains the deviation from values predicted by T0 alone; and stochasticity explains the deviation of observed T1 and T2 from values predicted by asymmetry. Right: Density distributions showing increasing variance as more components are added to doubling time estimates. **c** As the slopes of linear models between T0 and T1 or T2 are shallower than the identity line, stable points arise at the intersection between these lines and identity. **d** Graphical representation of T2 lineages converging toward the predicted equilibrium through the continuous inheritance of old poles. **e** Most of the data points concentrated around the predicted attractors, as shown by ellipse-like confidence regions ranging from 15% to 95% confidence. **f** Doubling times of old lineages from the mother machine shown over 24 h, as 13 independent lineages ($n = 745$ cells) remain stable over time. **g** Progression of a randomly chosen old lineage around the predicted equilibrium, which behaves as an attractor for the dynamic trajectory of the lineage over 60 generations

sections, we further analyze the patterns emerging from deterministic sources in our populations.

**Cell lineages are predicted to achieve growth equilibrium**. To investigate stable patterns arising from lineages in the mother machine, we investigated predictions observed in the phase plane. The regression lines of new and old subpopulations displayed slopes shallower than the identity line, indicating that conditions T1 = T0 or T2 = T0 must exist where each linear model crosses identity (Fig. 3c). This intersection is predicted to be a stable point[27], to which lineages converge by inheriting either pole consecutively over generations (Fig. 3d). Importantly, this property does not depend on a linear relationship between doubling times of a mother and its daughters. The same prediction arises from nonlinear relationships (Supplementary Fig. 3), provided that the slope at the intersect with identity is less than 1. Thus, bacterial populations are expected to stabilize around two equilibrium points where the doubling time of the daughter equals the doubling time of the mother, or T1 = T0 = 21.4 and T2 = T0 = 23.1 min. Old and new subpopulations from the mother machine concentrated around these predicted equilibrium values in the phase plane (Fig. 3e).

The verification of T1 and T2 equilibria requires long-term lineage data. The longest old daughter lineages were obtained from the mother machine, as old daughters remain trapped at the bottom of growth wells (Fig. 1c) for the length of the experiment. Our results revealed that the doubling times of old lineages remained remarkably stable over 53–60 generations (Fig. 3f and Supplementary Data 2), which is expected in the presence of equilibrium. By following lineages in the phase plane, we observed that doubling times consistently cycled around an equilibrium value of 23.1 min, despite stochastic patterns being also present (Fig. 3g). Thus, these results suggest the existence of a T2 equilibrium. As new daughters are only present in the mother machine traps for two generations, new lineages from this device were too short to verify the T1 equilibrium.

**Stable equilibria are connected by aging and rejuvenation**. Although the mother machine provided data for a strong characterization of the T2 equilibrium, it could not harbor enough new daughter divisions to verify the T1 equilibrium. Thus, we switched to the daughter device, which retains equally well both daughters (Fig. 2c, d). Following the approach used for characterizing the T2 equilibrium in the mother machine, we tracked T1 and T2 lineages inheriting the same pole consecutively over generations. The same general pattern seen in the mother machine was observed, except that two equilibria were now detected. Old and new daughter lineages achieved distinct equilibrium values of 20.6 and 19.5 min, respectively, displaying doubling time stability over time (Fig. 4a, Supplementary Data 3). On the phase plane, new and old lineages cycled around their respective equilibrium point (Fig. 4b), corresponding to the distinct physiological states produced by asymmetric division.

As the doubling times of old lineages remain centered around the T2 equilibrium, the mean value of old daughters in equilibrium does not manifest signs of progressive deterioration or aging over time (Figs. 3f and 4a). Instead, observing aging in a bacterial population requires that lineages are displaced from the T2 equilibrium. Such displacements will continuously occur when the population harbors both the T1 and T2 equilibria. When a new daughter at the T1 equilibrium becomes a mother, its new daughter remains at the same equilibrium, but its old daughter is now far removed from the T2 equilibrium (Fig. 4c). The old lineage generated from this daughter will converge over generations to the T2 equilibrium (Fig. 3d), displaying longer

doubling times with each division, thus manifesting aging. As a result, aging is predicted by the phase planes to be an integral part of bacterial populations if the T1 and T2 equilibria are present.

To test this prediction, we tracked lineages emerging from either equilibrium in the daughter device. These lineages belong to the same population as presented above, but following cells that are displaced from their correct equilibrium. We identified new daughters at equilibrium (Fig. 4b) and tracked a lineage of old daughters produced by these new daughters. We followed these old lineages over generations, and their trajectories confirmed a steady aging trend that connected the two equilibria (Fig. 4d, e, Supplementary Data 4, Supplementary Table 4). The transition from one equilibrium to the other required about three generations (Supplementary Fig. 4). Similarly, we followed new daughter lineages emanating from the T2 equilibrium as they converged toward the T1 equilibrium. As this convergence was now downwards, doubling times decreased with each generation. Thus, the emanating new lineages were rejuvenated (Fig. 4d, e), connecting the two equilibria in the opposite direction. The same trend shown in Fig. 4d was observed when expanding our analysis to five generations (Supplementary Fig. 4).

We tested whether these observations corroborated the characterization of the equilibria as stable attractors. By analyzing the doubling time distributions of each generation presented in Fig. 4d, we followed lineages leaving the opposite equilibrium and approaching their own. Following new lineages originated from the T2 equilibrium for three generations, we observed that these doubling time distributions varied significantly (Fig. 5a; one-way analysis of variance (ANOVA), F = 17.69, p < 0.001). A post-hoc analysis confirmed our observation that each new daughter distribution was increasingly displaced from the T2 equilibrium (Supplementary Fig. 4). This displacement happens gradually, with each generation displaying doubling times closer to the T1 equilibrium. In fact, the predicted equilibrium value of 19.5 min represented the true mean of the third new daughter generation OONNN (one-sample t-test, t = 0.512, p = 0.610), but not of its mother or grandmother (OONN and OON) (Supplementary Table 5). This structure was also verified for old lineages leaving the T1 equilibrium towards the T2 stable value of 20.6 min (Fig. 5b, Supplementary Fig. 4, and Supplementary Table 5; one-way ANOVA, F = 12.88, p < 0.001). These analyses suggest the presence of a structured doubling time increase or decrease as cells transition between equilibria through processes of aging and rejuvenation.

The convergence of the emanating lineages reconfirmed the stability of the system. Our results showed that, besides representing equilibrium states for long-term lineages, the T1 and T2 equilibria also behave as attractors that serve as targets for displaced lineages.

**Maintenance of equilibrium in the presence of stochasticity**. To verify the stability of equilibrium attractors in the presence of stochasticity, we performed a mathematical analysis based on long-term old lineage data (Fig. 3). For mother machine lineages, which represent the longest observation of the old lineage equilibrium, we had shown that stochasticity represents 78% of the doubling time variance. Although our deterministic factors predict stability (Fig. 3 and Supplementary Fig. 3), we considered whether stochasticity would prevent cell lineages from remaining in equilibrium over generations. The complete mathematical analysis is provided in Supplementary Note 1.

The combined effect of maternal doubling times and asymmetry estimate old daughter doubling times through a linear model of slope a = 0.347 and intercept b = 15.091 (Fig. 3a). As the slope predicted by these deterministic sources

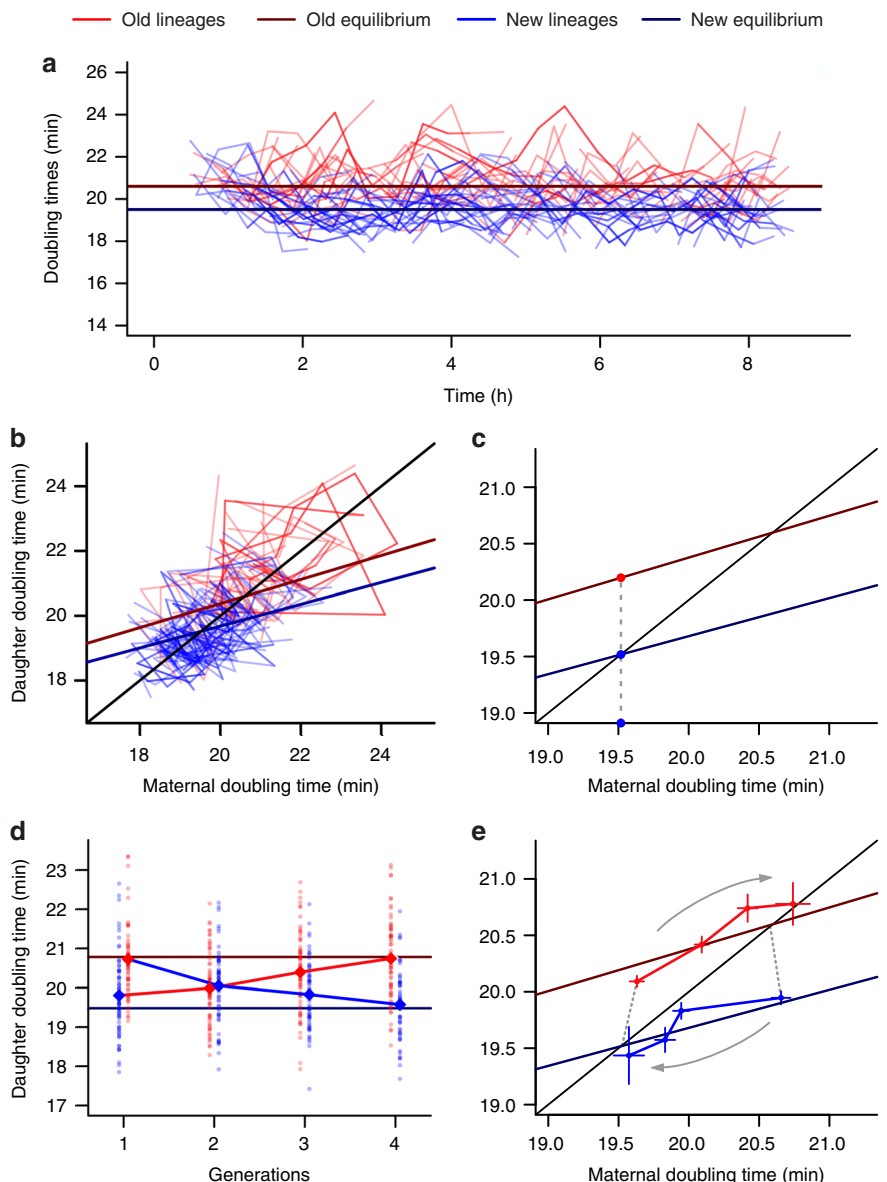

**Fig. 4** New and old daughters reach equilibrium in the daughter device. **a** New ($19.365 \pm 0.907$, $n = 158$) and old lineages ($21.177 \pm 1.237$, $n = 116$), represented by those inheriting the same pole for at least 4 generations, exhibited stably distinct doubling times ($t = 13.354$, df $= 200.66$, $p < 0.001$, one-tailed $t$-test) over time. **b** By following these lineages on the phase plane, we determined the equilibrium points based on the linear relationships between T0 and T1 or T2. A two-way ANCOVA revealed a significant effect of T0 ($F = 460.3$, $p < 0.001$) and age ($F = 334.2$, $p < 0.001$) on daughter doubling times ($n = 2400$), with both linear models displaying similar slope ($F = 1.051$, $p = 0.305$). **c** Graphical representation of two daughters born from a mother at T1 equilibrium. Although its new daughter remains at equilibrium, its old daughter displays a longer doubling time. **d** Old lineages born from T1 equilibrium (generation 1) converge towards T2 equilibrium over three generations. The same is observed for new lineages born from T2 equilibrium, which converge towards T1 equilibrium over time. **e** In the phase plane, we can observe the trajectory of cells born from the opposite equilibrium as they consecutively approach the equilibrium that matches their polarity. Error bars represent mean ± SEM. Old lineage $n = 370, 179, 70, 30$; new lineage $n = 291, 139, 62, 24$ cells

is shallower than 1, we observed the attractor T0 = T2 = 23.12 min. However, stochasticity could produce the variability observed in T2 by acting on either $a$ ($\sigma_1$) or $b$ ($\sigma_2$) (Supplementary Fig. 4). In this case, T2 = T0*($a + \xi_1$) + $b$ + $\xi_2$, where $\xi_1$ and $\xi_2$ are random variables drawn each generation from a Gaussian distribution with SD of $\sigma_1$ and $\sigma_2$, respectively. $\sigma_1$ could originate from stochasticity in processes such as asymmetric damage partitioning. A large $\sigma_1$ could lead to a continuous displacement of the old lineage if $a^2 + \sigma_1^2 \geq 1$, thus not allowing for stabilization (Supplementary Note 1). $\sigma_2$, on the other hand, represents an additive source of noise that does not disrupt stability.

To estimate the values of $\sigma_1$ and $\sigma_2$, we considered the linear model for our mother machine old lineages. If we assume stochasticity to act only on the slope $a = 0.374$, then $\sigma_1 \neq 0$ and $\sigma_2 = 0$. We can estimate $\sigma_1$ by calculating the deviation from $a$ for slopes calculated for each experimental pair of T0 and T2 in equilibrium, or (T2 – $b$)/T0 = $a + \sigma_1$. By performing this calculation, we obtained $\sigma_1 = 0.07$. The opposite scenario where $\sigma_1 = 0$ and $\sigma_2 \neq 0$ is also possible, in which case T2 – T0*$a$ = $b + \sigma_2$. In this case, our experimental doubling times indicated that $\sigma_2 = 1.62$. A biologically realistic scenario would likely exhibit stochasticity in both $\sigma_1$ and $\sigma_2$, resulting in $0 < \sigma_1 < 0.07$ and $0 < \sigma_2 < 1.62$. Thus, $\sigma_1 = 0.07$ represents the maximum stochasticity

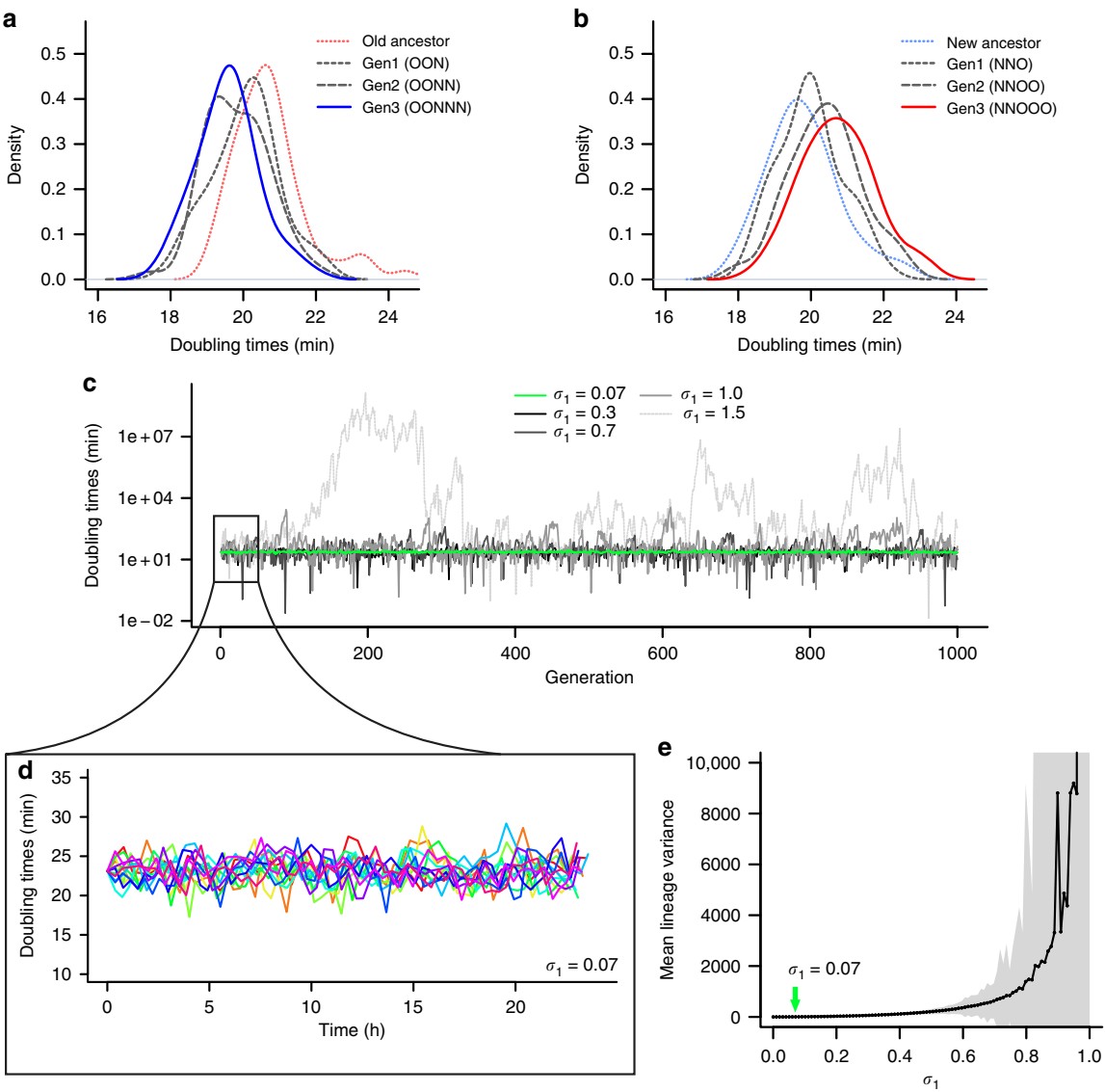

**Fig. 5** Doubling time equilibria have attractive properties for displaced and stable lineages. **a**, **b** Displaced lineages from the daughter device. **a** The doubling time distribution of a new lineage born from an old ancestor at the T2 equilibrium shifts towards shorter doubling times with each generation. After three generations of rejuvenation, the distribution becomes centered at the T1 equilibrium. **b** The opposite pattern appears from old lineages generated from the new equilibrium, which converge towards the T2 attractor over generations (Supplementary Table 5). **c**, **d**, **e** Evaluation of stochasticity in silico based on mother machine data (Fig. 3). **c** Doubling times of in silico old lineages starting from the T2 equilibrium, for increasing values of $\sigma_1$ in T2 = T0*$(a + \sigma_1) + b + \sigma_2$ (Supplementary Note 1). Stochasticity disrupts stability if $a^2 + \sigma_1^2 \geq 1$. **d** Doubling times generated for $\sigma_1 = 0.07$, the maximum stochasticity expected from our experimental data. **e** Mean doubling time variance of in silico old lineages followed for 100 generations (5000 replicates) for a range of $\sigma_1$ values (0.0001 to 1). When $\sigma_1 > 0.94$, the stability requirement $a^2 + \sigma_1^2 < 1$ is no longer fulfilled. Shaded area represents SD

that could influence the stability of our population. This value satisfies the condition for stabilization where $a^2 + \sigma_1^2 < 1$, for $0.347^2 + 0.07^2 = 0.125$. Therefore, the old lineage equilibrium observed in Fig. 3 behaves as a stable attractor.

We further tested this assertion by estimating the doubling times of simulated old lineages for different starting of $\sigma_1$ (Fig. 5c, d). Although an increase in stochasticity dramatically disrupts the stability of T2 over time, $\sigma_1 = 0.07$ reproduces the pattern observed in our data (compare Fig. 5d to Fig. 3f). As lineages maintain stability when $a^2 + \sigma_1^2 < 1$, equilibrium will be present for $\sigma_1 < 0.94$. The disruption of stability is easily visualized in Fig. 5e, where we simulated doubling times for a range of $\sigma_1$ values, following old lineages for 100 generations (5000 replicates for each $\sigma_1$). As $\sigma_1$ approaches 0.94, the doubling time variance increases sharply, indicating the loss of stabilizing properties. In

this scenario, old daughter doubling times would become increasingly large, which in real bacteria would mean the arrest of growth and proliferation. As Fig. 5e demonstrates, this threshold occurs for a much larger stochasticity than observed in our experiments.

Taken together, the data from this study indicates that bacterial populations displaying immortal proliferation reach states of physiological equilibrium for new and old lineages. These equilibria behave as attractors, to which displaced lineages constantly converge through aging and rejuvenation. Moreover, the equilibria display stabilizing properties despite the presence of stochasticity in the system. Therefore, deterministic patterns of stability connected by constant aging and rejuvenation emerge from bacterial populations in the presence of biological stochasticity.

## Discussion

In this study, we showed that the asymmetric partitioning of damage by mother bacteria explains the phenotypic distinction between old and new bacterial daughters, resulting in a landscape of aging and rejuvenation due to the transmission of damage down the daughter lineages. This landscape, quantified in the absence of stress or starvation, is visualized on a phase plane showing the doubling times of the old and new daughters as a function of the doubling time of the mother (Fig. 3a). We found that the doubling times of new and old lineages converged to distinct equilibrium values, T1 and T2, respectively, where they remained stable over time. Doubling times at the T1 equilibrium are shorter due to the inheritance of lower damage loads by new daughters. However, many cells within a mixed population exhibit doubling times that do not reside at equilibrium. For example, when a new daughter in equilibrium divides, it produces both a new and an old daughter. Although the new daughter doubling time remains at the T1 equilibrium, the old daughter is born at a distance from the T2 equilibrium (Fig. 5c). As this old daughter reproduces, the lineage it creates converges onto the T2 equilibrium (Figs 3c and 5d). During the convergence, it experiences increasing doubling times and aging. Reproduction by old daughters in equilibrium, likewise, produces lineages of new daughters that experience rejuvenation through decreasing doubling times, as they converge to the T1 equilibrium (Fig. 5d). Thus, the equilibria are stable attractors at which equilibrium lineages remain and to which displaced lineages converge. It is the behavior of these lineages, which emanate from one equilibrium and converge onto the other, that drives the dynamics of constant aging and rejuvenation in bacterial populations.

We interpret these equilibria to result from the opposing effects of aging and rejuvenation. All bacterial cells experience aging and rejuvenation as they grow and divide. Aging is driven by the acquisition of damage through the amount a cell receives from its mother and the de novo amount it accumulates during its lifetime[10,28]. This maternal contribution is evidenced by the positive slope for the regression of the doubling times of daughters onto their mothers (Fig. 3a). Mothers with longer doubling times generate daughters that also take longer to divide, presumably because these mothers transmit larger damage loads. This relationship is the bacterial version of the Lansing effect of rotifers, in which the offspring of older parents have shorter lifespan than the offspring of younger[29,30]. Rejuvenation results from the dilution of damage caused by the synthesis of new and damage-free materials by a cell, as indicated by the inheritance of a newly synthesized cell pole. The two stable equilibria are achieved when the dilution and acquisition of damage balance each other, such that a lineage allocates the same amount of damage from mother to daughter every generation. As asymmetric partitioning allocates less damage to new daughters, dilution is a stronger factor in these cells, resulting in a shorter doubling time at equilibrium. For the same reasons, new daughters produced by old daughters in equilibrium experience a sudden increase in dilution, displaying doubling time rejuvenation as they approach the T1 equilibrium. On the other hand, because old daughters receive larger damage loads by asymmetric partitioning, they experience the opposite of new daughters. Rejuvenation also demonstrates that the distinct physiological states of new and old daughters are not produced by mutations. If mutations accounted for this difference, new daughter lineages produced by the T2 equilibrium could not be rejuvenated.

The observation of large bacterial populations comprising both the T1 and T2 equilibria is essential for understanding the processes of bacterial aging and rejuvenation. The critical role of the two equilibria is demonstrated by the effect of witnessing only one equilibrium, through the hypothetical case of a symmetrical bacterium (Fig. 3b). With symmetry, identical old and new daughters are produced upon division and the population stabilizes around a single equilibrium. As a result, any observed variation of doubling times around equilibrium is properly attributed to stochastic noise, rather than aging and rejuvenation[31]. A similar observation derives from the observation of the T2 equilibrium in the mother machine. Although this design was extraordinarily innovative for the study of bacteria in physiological steady states[19], the stability of the single T2 equilibrium, when viewed in isolation (Fig. 3d, e), can give the impression that bacterial aging does not occur. Without a T1 equilibrium to create emanating lineages converging to T2 equilibrium, the cells in a mother machine appear to behave as symmetrical bacteria that do not age.

The emergence of age structures in bacterial populations through lineages of old and new daughters brings a new perspective to traditional views of biological aging. Although the progressive functional decline in old-pole bacteria comes to a halt once the lineage reaches equilibrium, we anticipate that an increased damage accumulation induced by extrinsic stress could result in a continuous deterioration leading to mortality—a more traditional definition of aging. In fact, the inheritance and accumulation of non-genetic damage is associated with aging in both bacteria[6–8] and traditional cellular systems[15,22,32], which could propose a unifying cause for aging at the cell lineage level. By redefining the old daughter as the continuation of the mother[4,5,33], bacterial replication can be seen as an individual mother bacterium budding off new daughters, thus retaining damage to produce rejuvenated individuals. In comparison with budding yeast[34] and the bacterium C. crescentus[3,33,35], the rejuvenated new daughters of E. coli represent a physiological juvenile, although morphologically indistinguishable from the mother. With the evolution of a distinction between a mother and a juvenile state in bacteria and other systems, partly due to asymmetric damage partitioning, age structure emerges in the population. As a juvenile daughter ages, its different life stages could experience different ecology and selection pressures. Future explorations remain necessary to determine the resilience of these age structures in face of such extrinsic pressures. Taking these notions together, bacteria could serve as a model for the evolutionary origins of aging, providing quantifiable long-term data on cellular aging and rejuvenation. Although aging in bacteria and traditional organisms will always have their differences, it may be that some key features of biological aging arose with the first microbes.

## Methods

**Bacterial strains and growth conditions**. All experiments were performed with K-12 E. coli wild-type strain MG1655. Before each experiment, cultures were inoculated in Luria-Bertani medium (LB broth; per liter: 10 g tryptone, 5 g yeast extract, 5 g NaCl; Sigma-Aldrich) and grown overnight at 37 °C with agitation. For culturing within microfluidic devices, the medium was supplemented with 0.075% Tween 20 to prevent the formation of biofilms.

**Microfluidic device design**. Two designs of microfluidic devices were used in this study. The first was based on the original mother machine design[19], modified for the inclusion of more flow channels and kindly provided by Ryan Johnson (University of California, San Diego). This design consisted of 16 flow channels bearing 2000 wells ($1.25 \times 30 \times 1$ m) each. When loaded into the device, bacteria enter the growth wells and remain trapped at the closed edge, consecutively inheriting old poles throughout the experiment. The wells comported up to seven cells at a time, before cells were washed into the large flow channel. The second design, here called the daughter device, was originally designed for the study of genetic oscillators[25] and comprises 48 growth chambers ($40 \times 50 \times 0.95$ μm) distributed in four columns; the chambers are flanked by 10 μm wide channels that provided fresh culture medium to bacteria throughout the experiment. For both devices, master silicon wafers were used as negative molds for the construction of polydimethylsiloxane microfluidic chips. Each soft lithography process yields 8–12

devices, which are attached to $24 \times 40$ mm coverslips through a Si–O–Si covalent bond, after exposure to $O_2$ and UV light.

**Cell loading and experimental conditions**. For loading the devices, overnight grown cultures were centrifuged for 2 min at 5300 g and supernatant media discarded, followed by pellet resuspension in 50 µL LB-Tween 20 medium. Microfluidic devices were placed in a vacuum chamber for 10 min and then loaded with bacteria by laying 3 µL of culture over the loading port. After verifying successful filling of the channels, input and output 60 ml syringes, containing 30 ml of culture medium and 10 ml of MilliQ water, respectively, were attached to the ports. The medium inlet was refilled as needed for the length of each experiment. All experiments were performed at 37 °C with constant supply of growth medium to ensure stable growth conditions. Replicates were performed in four independent microfluidic devices, and imaging began immediately after loading bacteria into the chambers.

**Time-lapse image acquisition**. Cell movies were collected by a Nikon Eclipse Ti-S microscope, with imaging intervals controlled by NIS-Elements AR software. Phase imaging of mother machine devices followed 2 min intervals, whereas 20 s intervals were used for the daughter device to ensure the correct tracking of all lineages.

**Image analysis for the quantification of bacterial growth**. Images were analyzed with the free software ImageJ (NIH, https://imagej.nih.gov/ij). By following microscopy images over time, we obtained growth and position information for each individual cell present in the field of acquisition. Cell coordinates were recorded as regions of interest (ROI) and annotated according to pole inheritance. From the ROIs, we determined the cell centroids, its length immediately before and after division, and the interval between cell divisions. Elongation rates (r) and corresponding doubling times ($\ln(2)/r$) were calculated from the data, and from the polarity annotation we determined maternity, sibling pairs and lineage trees. To ensure that the measurements were unbiased, we also performed blind data collections where cell length and time of division were recorded without previous knowledge of asymmetry and pole inheritance.

**Doubling time analysis**. Statistical analysis was performed using the software R version 3.4.1[36]. $p$-values < 0.05 were considered statistically significant. Sample sizes were determined according to previous studies in microfluidic devices[19], which reported doubling times of old and new daughters in the mother machine for the strain MG1655. Doubling times were recorded as defined above, and statistically analyzed without data transformations or corrections. Doubling times of new-old sibling pairs were compared through paired t tests as indicated in the figure legends. When analyzing doubling time relationships in the phase plane, linear regressions were performed between T0 and T1, or between T0 and T2. Two-way ANCOVA were performed to evaluate the effect of T0 and age (new or old) over T1 and T2. From the linear regressions, we determined the equilibrium points as the intersection between each linear model and the identity line, thus indicating a stable point where T0 = T1 or T0 = T2. Sample sizes (individual cells) are indicated along with reports of statistical analyses, usually located in the figure legends.

**Determination of cell positioning within the daughter device**. Centroids obtained for each cell at birth were evaluated according to their distance from the meridian of the growth chamber (horizontal axis). The chambers are open on both sides along the vertical axis; therefore, only the localization along the horizontal axis is relevant for starvation analyses.

**Partitioning the sum of squared deviations**. We partitioned the variance present in our data according to deterministic and stochastic sources. Using the sum of squared deviations method, the total variability ($SS_T$) of daughter doubling times was determined as the sum of T1 and T2 deviation from the population mean doubling times, or

$$SS_T = \sum_{i=1}^{n} \left(T1_i - \bar{T}_i\right)^2 + \sum_{i=1}^{n} \left(T2_i - \bar{T}_i\right)^2 \quad (1)$$

The first component of the total variance comprised the variability produced by the positive relationship between T0 and pooled T1 and T2, described by the line equation

$$\hat{T}_i = 0.283 \times T0 + 15.897 \quad (2)$$

The deviation of values predicted by the equation above from the mean daughter doubling times thus represent the variability introduced by the effect of a mother on its daughters:

$$SS_M = 2 \times \sum_{i=1}^{n} \left(\hat{T}_i - \bar{T}_i\right)^2 \quad (3)$$

Because new and old subpopulations are best described by individual linear models rather than a single one, variability is also introduced by asymmetry. We determined this component as the deviation of predicted T1 and T2 from the central

line in eq. (2):

$$\widehat{T1} = 0.219 \times T0 + 16.703 \quad (4)$$

$$\widehat{T2} = 0.347 \times T0 + 15.091 \quad (5)$$

$$SS_A = \sum_{i=1}^{n} \left(\widehat{T1}_i - \hat{T}_i\right)^2 + \sum_{i=1}^{n} \left(\widehat{T2}_i - \hat{T}_i\right)^2 \quad (6)$$

The last component of the total variability is determined as the deviation due to stochasticity, estimating the level of noise in doubling times. It is defined as the deviation of observed T1 and T2 from the values predicted by asymmetry, or

$$SS_S = \sum_{i=1}^{n} \left(T1_i - \widehat{T1}_i\right)^2 + \sum_{i=1}^{n} \left(T2_i - \widehat{T2}_i\right)^2 \quad (7)$$

Thus, by combining the sum of squares deviations, the fraction of the variation explained by deterministic factors is given as $(SS_M + SS_A)/SS_T$, and the fraction explained by stochasticity is determined as $SS_S/SS_T$.

**Analysis of stability in the presence of stochasticity**. The values of $\sigma_1$ and $\sigma_2$ were calculated from experimental doubling times of old lineages in equilibrium, grown in the mother machine device. For each pair of mother (T0) and daughter (T2), the effective slope was calculated as $(T2 - b)/T0 = a + \xi_1$, where $a$ and $b$ were obtained from a linear regression. $\sigma_1$ was calculated as the SD of $\xi_1$. A similar approach was performed for $\sigma_2$, given $T2 - T0 \star a = b + \sigma_2$. The resulting values were validated from an exploration of the parameter space for $T2 = T0 \star (a + \xi_1) + b + \xi_2$, where $\xi_1$ and $\xi_2$ are random variables drawn each generation from a Gaussian distribution with standard deviation of $\sigma_1$ and $\sigma_2$, respectively. To obtain a combination of $\sigma_1$ and $\sigma_2$ that satisfied our data, we randomly sampled $\xi_1$ and $\xi_2$ with a range of non-negative SDs. These values were used to predict T2 from T0, $a$ and $b$, looking for combinations of $\sigma_1$ and $\sigma_2$ that minimized the difference between variances of observed and estimated T2. This analysis yielded a range of values from $\sigma_1 = 0.07$, $\sigma_2 = 0$ to $\sigma_1 = 0$, $\sigma_2 = 1.62$, same output as mathematically predicted above.

## Data availability

The data supporting the findings of this study are available within the article and supplementary information.

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

## Acknowledgements

We thank M. Vergassola for assistance with the analysis of stability and stochasticity, and the reviewers for greatly improving this work with their suggestions. We thank R. Johnson and G. Graham for assistance with experimental setup, and S. Cheung, J. Chen, and A. Qiu for data assistance. Work was supported by grants to L.C. from the National Science Foundation (DEB-1354253) and funds from Donald Helinski. A.M.P. is supported by the Science Without Borders Fellowship/CAPES—Brazil, and by the Christopher Wills Graduate Student Research Award.

## Author contributions

A.M.P., C.U.R. and L.C. conceptualized the experimental design and methodology. A.M.P., C.B. and C.S. collected and validated experimental data. A.M.P., C.U.R., and L.C. analyzed the data. A.M.P. and L.C. wrote the paper. C.U.R. and L.C. provided supervision.

## Additional information

**Competing interests:** The authors declare no competing interests.

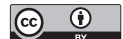 

