## [Peer Review File · Nature Communications]

Reviewers' comments:

Reviewer #1 (Remarks to the Author):

Whether aging exists even in unicellular microbes is a classical and important problem in microbiology. While aging in asymmetrically dividing bacteria is accepted, it still remains controversial whether symmetrically dividing bacteria such as *E. coli* also show aging under rich and constant environmental conditions. A previous study that tracked single-cell lineages on agar pads suggested the existence of aging in *E. coli* (Stewart, et al. PLoS Biol, 2005), but another study that tracked old-pole cell lineages for hundreds of generations with mother machine demonstrated the stability of elongation rate, which suggested the absence of aging (Wang, et al. Curr Biol, 2010).

Manuscript by Proenca et al. presents a unique insight that may resolve these seemingly contradicting results. The authors present that an *E. coli* cell produces physiologically asymmetrical daughter cells at cell division, and the doubling times of old-pole cell lineages and of new-pole cell lineages are progressively relaxed toward the distinct equilibrium states (T1 and T2) on average. The use of the two types of microfluidic device, mother machine and daughter device, is a unique and smart approach for evaluating the physiological differences between old-pole and new-pole cell lineages. I however think there are several major technical concerns for accepting their claims.

1. In this study, it is necessary to detect very small differences (~1 min) of mean doubling times in the presence of much greater phenotypic noise (stochasticity) of individual cells. However, the statistics in this study seem to ignore any measurement errors in estimating individual cells' doubling times. The authors should evaluate the errors that could be introduced in estimating doubling times, and justify that these errors are negligible for distinguishing the tiny differences of doubling times. According to the explanation in Methods, the authors first measured elongation rate (r) and determined the doubling time as $\ln(2)/r$. In the measurement of elongation rate, the information of birth length, division length and division interval was used. Therefore, the measurement errors of these parameters all matter to the estimation of doubling times. Precision of image analysis should affect cell size estimates. In the experiments with mother machine, time-lapse interval was set to 2 min; the estimate of division interval should contain about 10% error inevitably (assuming that the mean division intervals are about 20 min). Knowing these details of evaluating the individual cells' doubling times, I am afraid that the current data might not have sufficient statistical power to distinguish the differences.

2. Physiological states of cells are evaluated entirely by doubling times (elongation rates), but it is not apparent why this parameter is the most relevant to aging. For example, division interval, which is a similar but different physiological parameter from doubling time, would be important since it is directly related to the production of daughter cells. It would be informative to provide the results and discussion employing division interval as well and to examine whether the existence of two equilibrium points are independent of the choice of these physiological parameters.

3. In Fig 3A, the dependence of doubling times of old-pole and new-pole daughter cells on maternal doubling times is assumed linear. However, the experimental data points are scattered to a great extent, and linearity is not apparent; many different kinds of functional dependence could be assumed. The linearity assumption should be critical to determine the equilibrium doubling times T1 and T2 (Supplemental Table 2); the authors should justify the linear regression (or the precision of T1 and T2 values) from the experimental data more convincingly.

4. One of the important claims of this paper is that the doubling times of new-pole cell lineages and old-pole cell lineages are relaxed to different stable equilibrium points (T1 and T2) on average. Fig 5D

is a key result that supports this claim, but it is not shown whether the doubling times of the new-pole and old-pole cell lineages remain in these equilibrium points stably for longer generations. Daughter device should allow the tracking of these two different types of cell lineages longer (it is stated that the longest consecutive pole inheritance observed comprised 11 generations (L273)). The authors should prove the stability of these equilibrium points by extending the generations of Fig 5D.

5. I could not understand the advantages of using the measures such as 'distance to equilibrium', 'step size', 'return angle' and 'return probability' for analyzing the equilibrium. For example, 'distance to equilibrium' is defined as the Euclidean distance between mother-daughter point and equilibrium point on 2D plane (Supplemental Fig 1), but can it be simply the difference between the doubling time of a cell (T) and equilibrium doubling time, i.e. $|T-T_2|$, which would converge to 0 on average? The utility of the other parameters is also unclear to me. More explanations are required on the advantages of using these parameters.

In addition, it is not apparent to me whether the analyses done with those parameters (Fig 4 and 6) really support the claim as the experimental data points are scattered and deviated to large extents from the mean trends.

6. Though it might be more or less the issue of terminology, it is not clear to me whether the processes characterized in this study should be called as "aging" and "rejuvenation". The authors called the process of progressive increase of doubling times on old-pole cell lineages from the T1 equilibrium as "aging". However, the lineages reach the T2 equilibrium only in a few generations and stay there. Since the continuous deterioration of cellular functionality does not occur in this case, I wonder whether it is appropriate to call this process as "aging". For example, in Coelho, et al. *Curr Biol*, 2013, it is stated that "Aging is defined as slower reproduction and increased probability of death with time." The authors should clarify the definitions of "aging" and "rejuvenation" and explain the observed processes are in accordance with these definitions.

Minor points:

1. Abstract, L31-32, 'neither a damage-induced strategy': This is a misleading statement because some internal damage might be relevant to the observed aging behaviors as argued in Discussion.

2. Fig. 2(C): Some values of birth distance seem negative. Is this due to binning? If so, explain the detail on how the distributions are graphed.

3. In Methods, it is stated that stationary phase cells were loaded into the device and imaging started immediately after loading bacteria. How instantly did cells reach stable growth at the beginning of the time-lapse measurements? Did the authors discard the data from the first several generations to assure stable growth conditions?

4. Daughter device used in this study is essentially the same as microfluidic turbidostat reported by Ullman, et al. in *Phil. Trans. R. Soc. B.*, 2013. The authors must cite this paper and refer to it in the main text.

5. Supplemental Fig 1: If I understand the plots correctly, Mother and Daughter points in red are placed inappropriately because the y-value of Mother point should become the x-value of Daughter point. Is this correct?

Reviewer #2 (Remarks to the Author):

OVERVIEW

This manuscript investigates generational dynamics of *E. coli* aging in normal exponential growth, using doubling time as surrogate for physiological fitness. The authors tracked populations of dividing *E. coli* single-cell lineages in microfluidic devices designed specifically for such purposes. They confirmed the known effect of pole age: on average, inheriting the old pole has an aging effect on cells while the new pole in correlate with cellular rejuvenation. Through linear regression, the authors found partial correlations between successive generations of cells on either the elder or the younger branch of the family tree. They demonstrated graphically that the aging/rejuvenation effects of asymmetric division is attenuated through generations by this partial inheritance, resulting in two distinct stable points. They argue that these two phenotypic attractors provide "deterministic" age structures for the populations.

The core idea of the manuscript and the experimental design are novel and elegant. Stable age structure in bacteria could emerge from two opposing effects: phenotypic divergence through asymmetric divisions, and phenotypic convergence to stable attractors through partial inheritance. It was shown in previous studies [4,6,19] that for the first 7-10 generations, inheriting the old pole after each division reduces the growth rates incrementally. After this initial period the mother cell (always have the old pole) could maintain stable growth rates for up to 150 generations. The same paper [19] and subsequent studies [20] have demonstrated a stable and characteristic distribution for doubling time of *E. coli* single cells, and theoretical work [Pugatch 2015] connecting this distribution to physiological processes have been published.

The present manuscript attempts to resolve the apparent contradiction between robust growth and aging through asymmetric division, one of the central conceptual tensions in the field of aging, by investigating the dynamic landscape of phenotypic inheritance. In addition, the new microfluidic "daughter device" allow the authors to examine the generational effect of rejuvenation, and provided empirical results for a piece of the puzzle missing in the literature.

Taken together, the results described in this manuscript present a significant advance the field of bacterial aging in particular and holds much interest the general scientific public. The existence of aging in binary fissioning microbial such as *E. coli* is somewhat under-appreciated and conceptual challenging for audiences accustomed to organisms with more familiar life-histories. This study does have potential value of for the wider aging field, in demonstrating that an age structure could emerge from the common processes such asymmetric division and physiological homeostasis, despite non-traditional life-histories for the aging field.

However, the manuscript in the present form does not adequately support its core ideas due to drawbacks in data analysis, presentation and interpretation, esp. in the latter half of the manuscript (Fig.3 - Fig.6). The actual data for analysis - cell doubling times and lineage - are not available making any independent analysis to verify their findings in alternative modalities impossible. In conclusion, this study represents empirical advances in understanding bacterial replicative aging, and if the main results are better analysed and presented, it possesses a conceptual important advance worthy of publication in this journal.

MAJOR ISSUES

Our main concern with the manuscript, as far as our understanding goes (apologies if we missed

something...) are three interrelated main concerns with the data analysis and their graphical presentations and interpretation:

(i) The authors' contention that age structure in their data is deterministic involves circular reasoning, and is not clearly supported by the data. They use linear models to extract correlations between doubling times of successive generations, with majority of variations unexplained (the authors have not provided percentages of variation explained in any of their analysis). They then explain the mean population behaviour with deterministic dynamic systems consisting of extracted linear model parameters without considering the large error terms. The deterministic dynamic models built this way seem only useful in demonstrating the idea of dynamic attractors in a pedagogical sense. They then somehow concluded that this analysis supported a deterministic age structure, while the obvious stochastic nature of the data was prevented from entering the model in the first place.

An alternative explanation for the attractors would be statistical 'return to the mean' - drawing randomly at every division a doubling time from a given distribution is likely to draw a value close to the mean... We do not see any argument to favour a deterministic attraction to counter this argument (see also below).

(ii) The non-standard geometrical statistics described in Fig.4, Fig.6, and in supplementary figures are somewhat redundant, and statistically rather opaque. The quantities on x-axis or the y-axis are neither biologically meaningful, nor statistically independent from each other. For example, the maternal doubling time of a daughter is by definition the "daughter" doubling time of her mother, so the positioning of a pair of mother and daughter on the phase space is by definition already correlated. This particular problem is present in all 3 statistics used, but just take the "return probability" for instance, $d_2 < d_1$ simply means doubling time of daughter < doubling time of her grandmother. All these analysis seem to lead to one simple qualitative correlation for all populations involved: the further away a cell is from the mean doubling time, the more likely that her immediate progenies are closer to the mean than she is. It is misleading to use multiple related derivative statistics on the same data to give the impression of independent corroboration, while covering for the lack of statistical justification.

The authors are clearly aware of this problem, and attempt to address it through Monte Carlo simulation, albeit not in a convincing fashion. We are not convinced that the experimental data is different from the static Gaussian model in either Fig.4 or Fig.S2. The only statistical comparison the authors provided regarding this comparison is inappropriate: the sample points shown are not well capitulated by either the linear models or the static Gaussian model, and to test their parametric differences when they do not fit the data is meaningless. Moreover, even if the authors managed to reject the two simulated models, static Gaussian or random walks, they could not rule out other stochastic models with slightly different assumptions (Distributions other than Gaussian; random walks with dependent step sizes; etc...).

We find this approach (Fig.4, Fig.6, FigS1-2) based on geometries of the phase space statistically unjustified, and unworthy to pursue in further detail. Because additional statistical details further distracts away from the biological phenomena in question. We suggest that the authors present their data in a more biological meaningful manner, and test statistical hypothesis that is succinct and straightforward. A better analysis would consist of fitting the data to a parametric model that could capture both the mean behavior of the population and variabilities among the cells (see the next point). Then within the same family of models, test for parametric differences using likelihood ratio test, or test for the necessity of a deterministic term using tools such Akaike information criterion.

(iii) The presented analysis do not reject alternative models for the data. The problems we mentioned

above all are related to the fact that the authors chose to ignore the stochastic aspect of their data, and treating it as statistical noise and fit everything into the box of linear models. The closeness between the data and static Gaussian model (see above) hints at the relative strength of stochastic vs dynamic forces. Other approaches such as autoregressive models or more generally Markov models could much better integrate the stochastic and dynamic aspects of the data presented. In fact, using stochastic models to understand aging and life-history traits is actively field of research, including for microbes. [20, Steiner et al 2014] Stochastic models could capture other interesting aspects of the authors' data that is very relevant for the biological points they are making. For example, the auto-correlation and cross-correlation functions of doubling times through the generations (Fig.5D) could tell us how fast do the two stable points converge with each other.

MINOR ISSUES

(i) Abstract, line 31-32: The authors provided no evidence that the data is inconsistent with a damage-induced strategy. Molecular damages could arise stochastically in normal growing conditions.

(ii) Introduction is well-written but at times misrepresents the past advances in the field:

* Lines 43-44, 48-9: while protein damage localization to old poles was found to be correlated to fitness loss, no causal link was proven.

* Lines 50-51: "extrinsic damage", referred to in ref. 11 as resulting from fluorescence light excitation was already discarded in earlier work [6] where control experiment using only phase contrast light conditions.

* Lines 52-53: ref. 119 does show a transient decrease of growth rate in the first few generations as old pole cell. This effect is quickly attenuated, reaching a steady state (well examined in this current manuscript!).

* Lines 54-60: "starvation could be present in the most previous bacterial aging studies" is mis-stated. Earlier works (eg, [4,6] carefully controlled for homogeneous environment of excess nutrients by: (i) quantifying equivalent growth-rate within and among concentric 'onion shells' of single cells and (ii) comparing growth-rate locally of sister old-new pole cells sharing the same environment. Further, in agar-pad growth, old-pole daughters are invariably pushed to the extremity of the colony rather than found at the center. Stating this, the novelty of the new experimental setup in this manuscript is a welcomed advance and indeed corroborates the results obtained with the earlier technology of past decade.

(iii) Results

* Fig. 2A represents the doubling time difference between the OO, ON, NN, NO populations, probed (according to the legend) by a paired t-test. The actual distributions of deltas of each pair (eg old - new pole actual sisters) should be reported in Supplementary to demonstrate a mean value < 0 . This will assure no bias caused by outliers.

* Fig.2D: No statistical test of spatial homogeneity is provided; too many overlapping point to make any judgements visually.

* Fig.3A: Since the same mother cells divide to give rise a pair of old and new daughters, the effect of pole age needs to be tested comparing the pair to reflect their common heritage. As far as we could tell, this is done in Fig.2 as paired t-test, but not reflected in the linear regression models in Fig.3A. The red and blue lines seem to treat the old and new daughters as independent samples.

Compare Fig.3A-B to Fig.5B-C. The red populations in the two devices should be physiological similar, while the blue populations reflecting different stages of rejuvenation process. Does the slope being the same for the two populations in daughter devices while being different in mother machines have any biological interpretations?

* Fig.5B and Fig6A: Doubling time of the old lineage populations presented in these 2 figures seems to be much more variable than the new lineage populations, and may have long tail distributions. If true, it might suggest that the aging effect is more variable than rejuvenation, and distributed in non-symmetrical thus non-gaussian fashions. By not looking at the stochastic part of their data, the authors miss out important features of the biological effect that they are studying.

(iv) Methods

* The authors should ensure that they provided supplier information for all reagents and software used. For example, the LB components were purchased from which manufacturer? (e.g., Sigma Aldrich, St. Louis, MO, USA).

* details should also be provided for software including ImageJ and R.

* Raw data should be made available in Supplementary or deposited in a public databank. I

RECCOMENDATION

Considering the strength of novel ideas and solid experimental data, we recommend the manuscript to be considered for publication after major revisions., addressing the above issues.

ADDITIONAL BIBLIOGRAPHY

* Pugatch, R. Greedy scheduling of cellular self-replication leads to optimal doubling times with a log-Frechet distribution. Proceedings of the National Academy of Sciences 112, 2611-2616 (2015).

* Steiner, U. K., Tuljapurkar, S. & Coulson, T. Generation time, net reproductive rate, and growth in stage-age-structured populations. The American naturalist 183, 771-783 (2014).

Reviewers: Yifan Yang and Ariel Lindner

Reviewer #3 (Remarks to the Author):

I found the paper interesting and simple to follow. The experiments are cleverly designed, leading to robust results. My only suggestion is to perform few additional analysis, going beyond doubling times.

I found the discussion about the dynamics of doubling time very interesting. On the other hand, it is not clear to me what is driving that. The authors should connect that to previous literature on cell-size control. In particular, it is not obvious what is changing between old and new lineages, beyond doubling time. For instance: E. coli is known to have an adder size control. Is the added size changing between old and new lineages? Is the elongation rate changing? Do both new and old lineages have an adder behavior?

Said in other words, doubling time is an interesting quantity but is not the only worth to measure and, more importantly, duplication time alone, without information on the size and the growth rate, does not give the full story. Doubling time could be different because growth rates are different or because sizes are different or both.

RESPONSE TO REFEREES

We would like to thank the referees for the comprehensive analysis of our manuscript. We have made major revisions, and we believe that the manuscript has substantially improved from your suggestions. The main alteration consists on the addition of two new results sections, which focus on understanding the role of stochasticity in terms of variability and stability.

In the response below, we address major and minor comments point-by-point according the following format: **Comments from referees are bold and blue**. Response from the authors is in black, often including **"Citations from the revised manuscript"**.

RESPONSE TO REVIEWER #1:

Whether aging exists even in unicellular microbes is a classical and important problem in microbiology. While aging in asymmetrically dividing bacteria is accepted, it still remains controversial whether symmetrically dividing bacteria such as E. coli also show aging under rich and constant environmental conditions. A previous study that tracked single-cell lineages on agar pads suggested the existence of aging in E. coli (Stewart, et al. PLoS Biol, 2005), but another study that tracked old-pole cell lineages for hundreds of generations with mother machine demonstrated the stability of elongation rate, which suggested the absence of aging (Wang, et al. Curr Biol, 2010).

Manuscript by Proenca et al. presents a unique insight that may resolve these seemingly contradicting results. The authors present that an E. coli cell produces physiologically asymmetrical daughter cells at cell division, and the doubling times of old-pole cell lineages and of new-pole cell lineages are progressively relaxed toward the distinct equilibrium states (T1 and T2) on average. The use of the two types of microfluidic device, mother machine and daughter device, is a unique and smart approach for evaluating the physiological differences between old-pole and new-pole cell lineages. I however think there are several major technical concerns for accepting their claims.

We thank the reviewer for these positive remarks and detailed feedback. We have addressed the reviewer's concerns in detail below, indicating the modifications made to the original manuscript.

1. In this study, it is necessary to detect very small differences (~1 min) of mean doubling times in the presence of much greater phenotypic noise (stochasticity) of individual cells. However, the statistics in this study seem to ignore any measurement errors in estimating individual cells' doubling times. The authors should evaluate the errors that could be introduced in estimating doubling times, and justify that these errors are negligible for distinguishing the tiny differences of doubling times. According to the explanation in Methods, the authors first measured elongation rate (r) and determined the doubling time as $\ln(2)/r$. In the measurement of elongation rate, the information of birth length, division length and division interval was used. Therefore, the measurement errors of these parameters all matter to the estimation of doubling times. Precision of image analysis should affect cell size

estimates. In the experiments with mother machine, time-lapse interval was set to 2 min; the estimate of division interval should contain about 10% error inevitably (assuming that the mean division intervals are about 20 min). Knowing these details of evaluating the individual cells' doubling times, I am afraid that the current data might not have sufficient statistical power to distinguish the differences.

Indeed, the difference between new and old daughter doubling times is very small in unstressed conditions, especially within microfluidic devices. According to Wang et al. [19], who measured hundreds of thousands of cells in the original mother machine, we expected old daughters to display an average generation time of 21.06 min, and new daughters to divide every 20.14 min. Despite the small difference between these populations, a power analysis indicated that 532 pairs of new and old daughters would be enough to detect a distinction with a significance level of 0.05. Since our analysis included thousands of cells, and the old daughters displayed longer doubling times consistently throughout the study, we have shown that our data possessed statistical power to detect such physiological asymmetry.

With that said, the fact that our data presents great phenotypic stochasticity is an important concern. Stochasticity at the single-cell level is a key mechanism for the generation of phenotypic heterogeneity, which is advantageous for bacterial populations. We have previously addressed the relationship between stochasticity and determinism through computational methods (Chao et al., *PLoS Comput. Biol.* 2016 [31]), and we were pleased to include a section in the present manuscript to further expand our analysis. We partitioned the sums of squared deviations for our experimental doubling times to determine the fractions of the variability determined by maternal doubling times, asymmetry, and stochasticity:

“Despite following the trends described by linear regression, new and old subpopulations displayed large variance. To determine the sources of this variance, we partitioned the sums of squared deviations for T1 and T2 doubling times (see Supplementary Methods). We started by identifying the total sum of squares as the deviation of each doubling time from the population mean, obtaining $SS_T = 6737.66$. Some of this total variance was produced by the positive relationship between a mother and its daughters ($\beta = 0.28$; Fig. 3B, bottom panels). This fraction of the variance, SS_M , was determined as the deviation of predicted pooled T1 and T2 from the population mean doubling time, 22.45 min. Thus, the sum of squared deviations due to maternal inheritance was obtained as $SS_M = 557.76$. Due to asymmetry, however, doubling times predicted by two separate linear models deviated from the values predicted by the mother alone (Fig. 3B, middle panels). This deviation, produced by asymmetry, was determined as $SS_A = 926.32$. Finally, observed doubling times deviated from predicted values due to stochasticity, with $SS_S = 5253.58$ (Fig. 3B, top panels). By combining these values, the fraction of the variance determined by deterministic sources is given as $(SS_M + SS_A)/SS_T = 0.22$, while the remaining variance was explained by stochastic factors $SS_S/SS_T = 0.78$. Daughter device populations obtained similar results, with 24.9% of the variation explained by non-genetic maternal inheritance and asymmetry (Supplementary Table 3).”

We have also included Supplementary Fig. 1, which estimates the introduction of artificial error due to length measurements and compares growth parameters. Division intervals and doubling times expressed

the same stability over time and mean population values (20.06 ± 3.49 and 20.13 ± 1.11 min, respectively). However, contrary to introducing more noise in our sample, doubling times exhibited markedly lower variance (coefficient of variation = 5.53%) than division intervals (CV = 17.38%). This comparison was performed for daughter device measurements, with an image acquisition interval of only 20s. In the mother machine, where the larger acquisition interval could account for a greater disparity of division intervals, estimating growth asymmetry in terms of elongation rates (per minute) and corresponding doubling times becomes even more necessary.

2. Physiological states of cells are evaluated entirely by doubling times (elongation rates), but it is not apparent why this parameter is the most relevant to aging. For example, division interval, which is a similar but different physiological parameter from doubling time, would be important since it is directly related to the production of daughter cells. It would be informative to provide the results and discussion employing division interval as well and to examine whether the existence of two equilibrium points are independent of the choice of these physiological parameters.

We agree in recognizing the relevance of other physiological parameters for the study of aging. To address this concern, we have included a comparison of division intervals and our previous growth parameters in Supplementary Fig. 1. In unicellular systems such as *Caulobacter* and budding yeast, mother cells exhibit a decline in both division intervals and reproductive outputs over time. Unstressed *Escherichia coli* populations, on the other hand, were shown to display impressive stability over time [19]. Division intervals express this same stability in our study (Supplementary Fig. 1), despite much higher variance. The asymmetric physiology studied in our manuscript, however, is magnified by cell growth disparities (elongation) in combination with division interval differences. For this reason, reporting elongation rates and corresponding doubling times has been the standard procedure in the field [4,6,8,11,19].

3. In Fig 3A, the dependence of doubling times of old-pole and new-pole daughter cells on maternal doubling times is assumed linear. However, the experimental data points are scattered to a great extent, and linearity is not apparent; many different kinds of functional dependence could be assumed. The linearity assumption should be critical to determine the equilibrium doubling times T1 and T2 (Supplemental Table 2); the authors should justify the linear regression (or the precision of T1 and T2 values) from the experimental data more convincingly.

We have included a new section in the manuscript dedicated to partitioning the variance into deterministic and stochastic components. Stochasticity plays a large role in our data, which might originate from noise in gene expression, damage partitioning, etc. The use of linear regression was justified by comparing it to non-linear models through the Akaike information criterion (AIC; Supplementary Fig. 3A). All models show a similar relationship between doubling times of a mother and its daughters, and the AIC shows limited improvement by choosing a model over the other.

What is more important, however, is that the choice of model is not critical to determine the existence of stable equilibria. We have included graphical representations of non-linear models in Supplementary Fig.

3, showing that the stability arises whenever the slope of the model is shallower than the identity slope at the intersection with the latter. By choosing a point in the neighborhood of the intersection and applying the line equation repeatedly to predict the following generations, the lineage converges to the stable point. We have clarified this aspect of our prediction in the manuscript:

“To investigate stable patterns arising from lineages in the mother machine, we investigated predictions observed in the phase plane. The regression lines of new and old subpopulations displayed slopes shallower than the identity line, indicating that conditions $T1 = T0$ or $T2 = T0$ must exist where each linear model crosses identity (Fig. 3C). This intersection is predicted to be a stable point²⁷, to which lineages converge by inheriting either pole consecutively over generations (Fig. 3D). Importantly, this property does not depend on a linear relationship between doubling times of a mother and its daughters. The same prediction arises from non-linear relationships (Supplementary Fig. 3), provided that the slope at the intersect with identity is less than 1. Thus, bacterial populations are expected to stabilize around two equilibrium points where the doubling time of the daughter equals the doubling time of the mother, or $T1 = T0 = 21.4$ and $T2 = T0 = 23.1$ min.”

4. One of the important claims of this paper is that the doubling times of new-pole cell lineages and old-pole cell lineages are relaxed to different stable equilibrium points (T1 and T2) on average. Fig 5D is a key result that supports this claim, but it is not shown whether the doubling times of the new-pole and old-pole cell lineages remain in these equilibrium points stably for longer generations. Daughter device should allow the tracking of these two different types of cell lineages longer (it is stated that the longest consecutive pole inheritance observed comprised 11 generations (L273)). The authors should prove the stability of these equilibrium points by extending the generations of Fig 5D.

We agree that it is necessary to show that new and old lineages remain in equilibrium for generations. We begin our analysis of daughter device equilibria in Fig. 4 (former Fig. 5) by showing the long-term stability. We show that lineages inheriting the new or old poles for 5-6 consecutive generations display stable growth over time (Fig. 4A) and exhibit a cycling behavior around their predicted equilibria on the phase plane (Fig. 4B). Only afterwards we focus on the transition between equilibria happening in the same population (Fig. 4D and 4E). We have included a consideration of this points as “**These lineages belong to the same population as presented above, but following cells that are displaced from their correct equilibrium**”. We apologize that it was not clear in the original manuscript.

While it is true that our longest lineage had 11 generations, this referred to a single lineage. Continuous lineages are not common in the daughter device, which removes spatial age biases from the sample. For this reason, the sample size decreases dramatically for each generation added to the analysis. To show that the trend observed in Fig. 5D continues on, we have extended it for one more generation in Supplementary Fig. 4.

For further considerations of the behavior of lineages in equilibrium, we performed experiments in the mother machine presented in Fig. 3. We have also performed new analyses on the robustness of this equilibrium in the presence of stochasticity (see comment below).

5. I could not understand the advantages of using the measures such as 'distance to equilibrium', 'step size', 'return angle' and 'return probability' for analyzing the equilibrium. For example, 'distance to equilibrium' is defined as the Euclidean distance between mother-daughter point and equilibrium point on 2D plane (Supplemental Fig 1), but can it be simply the difference between the doubling time of a cell (T) and equilibrium doubling time, i.e. $|T - T_2|$, which would converge to 0 on average? The utility of the other parameters is also unclear to me. More explanations are required on the advantages of using these parameters.

In addition, it is not apparent to me whether the analyses done with those parameters (Fig 4 and 6) really support the claim as the experimental data points are scattered and deviated to large extents from the mean trends.

We agree with the reviewer that these parameters may not express a strong biological meaning, and a more robust analysis could be performed to evaluate the stability of attractors. To address this concern, which was also raised by reviewer 2, we have developed a new analysis of equilibrium stability in correspondence with Dr. Massimo Vergassola, UC San Diego. This is now presented in the concluding section of our results, "**Maintenance of stable equilibrium in the presence of stochasticity**".

Our new analysis consists of a mathematical evaluation of stability in stochastic cell lineages, where the insertion of noise has biological meaning tied to the deterministic aspect of the. We show that, for a linear model predicting doubling times of old daughters (T_2) from maternal values (T_0), stochasticity could act on either the slope or the intercept. Thus, $T_2 = T_0 \cdot a + b$ becomes $T_2 = T_0 \cdot (a + \sigma_1) + b + \sigma_2$:

"Stochasticity could produce the variability observed in T_2 by acting on either a (σ_1) or b (σ_2) (Supplementary Fig. 4). In this case, $T_2 = T_0 \cdot (a + \xi_1) + b + \xi_2$, where ξ_1 and ξ_2 are random variables drawn each generation from a Gaussian distribution with standard deviation of σ_1 and σ_2 , respectively. σ_1 could originate from stochasticity in processes such as asymmetric damage partitioning. A large σ_1 could lead to a continuous displacement of the old lineage if $a^2 + \sigma_1^2 \geq 1$, thus not allowing for stabilization (Supplementary Note 1). σ_2 , on the other hand, represents an additive source of noise that does not disrupt stability.

To estimate the values of σ_1 and σ_2 , we considered the linear model for our mother machine old lineages. If we assume stochasticity to act only on the slope $a = 0.374$, then $\sigma_1 \neq 0$ and $\sigma_2 = 0$. We can estimate σ_1 by calculating the deviation from a for slopes calculated for each experimental pair of T_0 and T_2 in equilibrium, or $(T_2 - b) / T_0 = a + \sigma_1$. By performing this calculation, we obtained $\sigma_1 = 0.07$."

This level of stochasticity, with $\sigma_1 = 0.07$, largely satisfied the stability requirement. In fact, for the experimentally obtained slope $a = 0.347$, stability will only be lost for $\sigma_1 > 0.94$ — a value much higher than we observed. We hope that this new mathematical evaluation will corroborate the stability of doubling time attractors observed in our experiments.

6. Though it might be more or less the issue of terminology, it is not clear to me whether the processes characterized in this study should be called as “aging” and “rejuvenation”. The authors called the process of progressive increase of doubling times on old-pole cell lineages from the T1 equilibrium as “aging”. However, the lineages reach the T2 equilibrium only in a few generations and stay there. Since the continuous deterioration of cellular functionality does not occur in this case, I wonder whether it is appropriate to call this process as “aging”. For example, in Coelho, et al. *Curr Biol*, 2013, it is stated that “Aging is defined as slower reproduction and increased probability of death with time.” The authors should clarify the definitions of “aging” and “rejuvenation” and explain the observed processes are in accordance with these definitions.

The definition of aging and rejuvenation is a relevant terminology issue, which we are happy the reviewer is bringing up. From the early studies, the field of bacterial aging has been guided by the broader definition — derived from the evolutionary theory — that aging is a decline in fitness with time. This definition relies on the retention of damage by a parent, in order to produce rejuvenated offspring. Old-pole cells essentially behave as a parent, retaining protein aggregates and producing new daughters with smaller damage loads. This definition is also accepted from a cellular biology perspective, considering a “time-dependent functional decline” the broad definition of aging [32]. However, many authors consider a *continuous decline* and *increasing probability of death* as essential hallmarks of aging. This discrepancy has led to diverging interpretation of past studies, with controversy still remaining on whether asymmetric bacteria, with its unique way of partitioning damage, undergo aging.

One of our goals with this study is to address this controversy. Our results show that old daughters exhibit a decline in growth rates (longer doubling times) and consequently reproduce at a lower rate than new daughters. While this deterioration comes to a halt after reaching equilibrium, more old daughters are continuously produced by new-pole cells in all stages of rejuvenation. Therefore, even though a single old lineage reaches equilibrium in a few generations, the population itself comprises continuous processes of aging and rejuvenation with every cell division.

In the present study we aimed at bacteria growing in the absence of extrinsic damage, to verify whether cells undergoing stable growth remained asymmetric. However, had we induced the accumulation of larger damage loads, it is possible that asymmetry would lead to an increasing probability of death in lineages with progressively longer doubling times. This would be consistent with studies such as Wang et al. (*Curr Biol*, 2010), which found no functional decline in unstressed populations, and Coelho et al. (*Curr Biol*, 2013) and Rang et al. (*Microbiology* 2012), who detected progressive deterioration in stressed microcolonies of yeast and *E. coli*. Thus, we anticipate that the probability of death in old lineages is linked to the accumulation of damage in the system.

We consider the unification of distinct subfields of aging research to be a pressing matter. As such, we hope that our concluding paragraph might conciliate some of the diverging views on the biology of aging:

“The emergence of age structures in bacterial populations through lineages of old and new daughters brings a new perspective to traditional views of biological aging. **Although the progressive functional decline in old-pole bacteria comes to a halt once the lineage reaches equilibrium, we anticipate that an increased damage accumulation induced by extrinsic stress could result in a continuous deterioration leading to mortality — a more traditional definition of aging.** In fact, the inheritance and accumulation of non-genetic damage is associated with aging in both bacteria^{6–8} and traditional cellular systems^{15,22,32}, which could propose a unifying cause for aging at the cell lineage level. (...)”

Minor points:

1. Abstract, L31-32, ‘neither a damage-induced strategy’: This is a misleading statement because some internal damage might be relevant to the observed aging behaviors as argued in Discussion.

Our intention was referring to the effect of extrinsic damage, which would induce a larger accumulation of internal damage. We apologize this was not clear. We have reworded the statement as **“neither a strategy induced by extrinsic damage”**.

2. Fig. 2(C): Some values of birth distance seem negative. Is this due to binning? If so, explain the detail on how the distributions are graphed.

That is correct, it was a binning effect of density distributions. We have edited the figure for clarity.

3. In Methods, it is stated that stationary phase cells were loaded into the device and imaging started immediately after loading bacteria. How instantly did cells reach stable growth at the beginning of the time-lapse measurements? Did the authors discard the data from the first several generations to assure stable growth conditions?

The first couple of generations tends to display slower growth, as they are still acclimating to the device. Because the polarity of the first cells arriving in the device is unknown, it takes three generations to recognize new and old daughters in our populations. Those first generations are not included in our analysis.

4. Daughter device used in this study is essentially the same as microfluidic turbidostat reported by Ullman, et al. in Phil. Trans. R. Soc. B., 2013. The authors must cite this paper and refer to it in the main text.

We have added this reference to our manuscript [26]. Our microfluidic design was developed by the Jeff Hasty Lab in 2011 (Mondragón-Palomino 2011, Science [25]) and seems to have inspired the device used by Ullman et al. as well.

5. Supplemental Fig 1: If I understand the plots correctly, Mother and Daughter points in red are placed inappropriately because the y-value of Mother point should become the x-value of Daughter point. Is this correct?

The reviewer is absolutely correct. Thank you. We have replaced the figure.

RESPONSE TO REVIEWERS #2:

MAJOR ISSUES

Our main concern with the manuscript, as far as our understanding goes (apologies if we missed something...) are three interrelated main concerns with the data analysis and their graphical presentations and interpretation.

Because the major three issues (i), (ii), and (iii), raised by Reviewers #2 all involve the questions of attractors, determinism, and stochasticity, we will address them in one combined discussion. We have asked our statistical and dynamical systems colleagues for suggestions on how best to address the rightfully raised issues. In particular, we consulted two experts in the field of stochastic processes in dynamical systems at UCSD (Professors George Sugihara and Massimo Vergassola). On the basis of those discussion, we have come to believe that the best way address Reviewers #2 is to divide the discussion into two parts, first when is “return to the mean” a factor and second what is the effect of stochasticity on our proposed deterministic and stable attractors. We argue that while return to the mean and attractor stability are equivalent when a lineage of cells is at the proposed equilibrium (see next sentences), return to the mean is not the explanation when the doubling times of cells are away from the equilibrium. We will then use an analysis provided by Professor Vergassola to show that, at the equilibrium, the level of stochasticity observed in our data is not sufficiently strong to **disrupt** the stabilizing deterministic factors. From a statistical perspective, the deterministic factors result from the regression of a daughter’s doubling time onto the mother’s doubling time (Fig. 3) with a correlation of less than 1, and hence return to the mean. From a dynamical systems approach, the deterministic factors result from the slope of the regression being less than one. Thus, return to the mean and dynamic stability are the same phenomenon in a two-dimensional system, and they are just different words used by two fields (statistics and dynamical systems) to describe the same thing. All of these issues, in addition to new data and analysis, have been added to the revised manuscript. Professor Vergassola’s analysis has been added to the Supplemental Information. We provide below a brief summary of our arguments.

1) Return to the mean is not occurring when away from the equilibrium. To address whether return to the mean operates when a lineage of bacteria is away from the equilibrium, we performed a new evaluation on the distribution of doubling times for lineages removed from equilibrium. The logic is that if the doubling times of distant cells were simply returning to the mean, the distribution of those cells should be the same as the distribution of cells already at the equilibrium. Our results showed that distant cells and equilibrium cells formed distributions that were statistically and significantly different (Fig. 5, Supplementary Fig. 4, and Supplementary Table 5). Cells that were at intermediate distances also formed a significantly different distribution. More importantly, as a lineage underwent cell divisions, the daughters formed distributions with means closer to the equilibrium than their mothers. Thus, return to the mean cannot explain the convergence of lineages to equilibrium. These results alone are strong, if not conclusive, evidence that the equilibria in our systems behave as stable attractors. The system is noisy and stochastic, as evidenced by

the distribution of doubling times at each step of the approach to equilibrium, but the level of stochasticity is clearly not sufficiently strong to destabilize the attractor. However, because Reviewers #2 requested additional treatment of the problem by a better “*parametric model that could capture both the mean behavior of the population and variabilities among the cells*”, we present next the analysis of Professor Vergassola (included as the last section of our results). We believe that this new analysis should capture the key relationship between the deterministic (mean) and stochastic (variance) behaviors of the cells.

2) Parametric model for the deterministic and stochastic behavior of cells. From our phase plot of the doubling times of daughters to mothers (Fig. 3 and 4B), we note that daughters and mothers are just two subsequent points in a time series, and use Professor Vergassola’s more general notation to represent them as $X(n+1)$ and $X(n)$. For a population of either old or new daughters, the deterministic relationship of the mean behavior is given by a linear regression,

$$X(n+1) = X(n) a + b \quad (\text{Eq1})$$

where $a = 0.3472$ and $b = 15.09$ were estimated from our data (old daughter regression line, Fig. 3A). NB: In response to Reviewer #1, our revised manuscript includes an analysis to show that a linear model gives a better fit than adding a higher order quadratic (Supplementary Fig. 3).

The conditions for Eq1 being stable is given by determining the changes at two time steps

$$\Delta X(n+1) = X(n+1) - X(n) \text{ and}$$

$$\Delta X(n) = X(n) - X(n-1) \quad (\text{Eqs2})$$

By combining Eqs2 and Eq1, we find that

$$\Delta X(n+1) = \Delta X(n) a \quad (\text{Eq3})$$

Thus, if $a < 1$, the next change in X is deterministically always smaller and X should converge to an equilibrium that is an attractor. Our estimate was $a = 0.3472$, thus smaller than 1. Note also that the equilibrium value (X^*) is given by Eq1 when

$$X(n+1) = X(n) = X^* \text{ and}$$

$$X^* = b / (1 - a) \quad (\text{Eq4})$$

The argument for stability was presented in our original manuscript, but only graphically (Fig. 3C and 3D). Eq1 is a better approach because it allows the explicit addition of stochasticity. Let σ_1 and σ_2 be the standard deviation of the variability of a and b , respectively. Thus,

$$X(n+1) = X(n)(a + \sigma_1) + (b + \sigma_2) \quad (\text{Eq5})$$

The addition of σ_1 and σ_2 has different effects on the stability of the attractor. σ_2 creates stochasticity by varying the experienced value of b (and hence also X^*) over time, but it does not affect the stability imposed

by Eq3. However, the magnitude of σ_1 can have a profound effect on stability in Eq3 and Eq5. If σ_1 is large, chance sampling could make $(a + \sigma_1) > 1$ sufficiently common to destabilize Eq3 and the variance of $X(n)$ is expected to increase over time. If σ_1 is small, a chance sampling of $(a + \sigma_1) > 1$ should be rare, chance deviations from X^* should decay, and the variance of $X(n)$ is bounded. Professor Vergassola obtained a numerical solution for the variance of $X(n)$ and found that the variance is bounded if $a^2 + \sigma_1^2 < 1$, or

$$\sigma_1 < \sqrt{1 - a^2} \quad (\text{Eq6})$$

3) What happens if Eq6 is satisfied? If Eq6 is true, the equilibrium behaves as a stochastic but stable attractor. Moreover, the variance created by stochasticity is bounded, in which case there is returning to the mean. We note however, as we showed (see point 1 above), returning to the mean is not the reason why the doubling times of the lineages far away from the equilibrium move to the equilibrium. The movement of the far away lineages results from the deterministic effects of Eq1 with the estimated slope of $a = 0.3472$.

4) What happens if Eq6 is not satisfied? If Eq6 is not true, the equilibrium is destabilized by stochasticity because the variance becomes unbounded. With no bounds, stochasticity can grow unchecked. Additionally, return to the mean is now not realized because we are not sampling from the same distribution over time.

5) Is Eq6 satisfied empirically in an experimental bacterial lineage at the equilibrium? To determine if our experimental data conformed to Eq6, we estimated σ_1 with Eq5 by from a sample of mother and daughter doubling times at equilibrium. By letting the mother and daughter doubling times represent $X(n)$ and $X(n+1)$ respectively, we solved for σ_1 by letting $\sigma_2 = 0$ (sample size is $n = 720$ pairs). By letting $\sigma_2 = 0$, we in fact estimated the maximum value of σ_1 . If the maximum value of σ_1 satisfies Eq6, then the empirical equilibrium is stable because letting $\sigma_2 > 0$ only reduces the value of σ_1 . Our estimated result was $\sigma_1 = 0.07$, which is greatly less than $\sqrt{1 - 0.3472^2} = 0.94$, and Eq6 is satisfied. We note that we also estimated a value of maximum $\sigma_2 = 1.66$ by letting $\sigma_1 = 0$. Our data currently lacks the power to estimate σ_1 and σ_2 simultaneously to get their realized values, which are both likely to be greater than zero.

To demonstrate the stabilizing effect Eq6, we have added simulated data of equilibrium lineages (Fig. 5C, D and E) to demonstrate that we can replicate the stability of the empirical data with $\sigma_1 = 0.07$ and that increasing σ_1 to the proximity of threshold $\sqrt{1 - 0.3472^2} = 0.94$ greatly increases variance.

5) How about moments higher than variance? Even if Eq6 is true and the second moment of variance is bounded and stabilized, it is possible that increases of a higher moment can destabilize the equilibrium. Professor Vergassola has derived the conditions for destabilization and finds that our estimate of maximal $\sigma_1 = 0.07$ is sufficiently small that the requirements for the 3rd and 4th moments are also easily met. As he concludes, the transition to moments that do not stabilize takes place for the order in the hundreds, which are likely not reached by any realistic measurements of a study such as ours.

In summary, we very much appreciate Reviewers #2 suggestions for implementing a more formal analysis of stochasticity and stability. We believe that Professor Vergassola's analysis demonstrates clearly that while the amount of stochasticity in our equilibrium lineages inflates the variance of doubling times, it is not sufficiently large to overcome the deterministic and stabilizing properties of the attractor.

(i) The authors' contention that age structure in their data is deterministic involves circular reasoning, and is not clearly supported by the data. They use linear models to extract correlations between doubling times of successive generations, with majority of variations unexplained (the authors have not provided percentages of variation explained in any of their analysis). They then explain the mean population behaviour with deterministic dynamic systems consisting of extracted linear model parameters without considering the large error terms. The deterministic dynamic models built this way seem only useful in demonstrating the idea of dynamic attractors in a pedagogical sense. They then somehow concluded that this analysis supported a deterministic age structure, while the obvious stochastic nature of the data was prevented from entering the model in the first place.

An alternative explanation for the attractors would be statistical 'return to the mean' - drawing randomly at every division a doubling time from a given distribution is likely to draw a value close to the mean... We do not see any argument to favour a deterministic attraction to counter this argument (see also below).

Please refer to general response above.

We have modified our manuscript to include a clear distinction between deterministic and stochastic components of the variability observed in our data, quantifying the contribution of each component. As described above, we have added a new analysis focusing on the stability of attractors in the presence of stochasticity. We believe the manuscript has greatly improved from following the Reviewers' advice.

(ii) The non-standard geometrical statistics described in Fig.4, Fig.6, and in supplementary figures are somewhat redundant, and statistically rather opaque. The quantities on x-axis or the y-axis are neither biologically meaningful, nor statistically independent from each other. For example, the maternal doubling time of a daughter is by definition the "daughter" doubling time of her mother, so the positioning of a pair of mother and daughter on the phase space is by definition already correlated. This particular problem is present in all 3 statistics used, but just take the "return probability" for instance, $d_2 < d_1$ simply means doubling time of daughter < doubling time of her grandmother. All these analysis seem to lead to one simple qualitative correlation for all populations involved: the further away a cell is from the mean doubling time, the more likely that her immediate progenies are closer to the mean than she is. It is misleading to use multiple related derivative statistics on the same data to give the impression of independent corroboration, while covering for the lack of statistical justification.

The authors are clearly aware of this problem, and attempt to address it through Monte Carlo simulation, albeit not in a convincing fashion. We are not convinced that the experimental data is different from the static Gaussian model in either Fig.4 or Fig.S2. The only statistical comparison the authors provided regarding this comparison is inappropriate: the sample points shown are not well capitulated by either the linear models or the static Gaussian model, and to test their parametric differences when they do not fit the data is meaningless. Moreover, even if the authors managed to reject the two simulated models, static Gaussian

or random walks, they could not rule out other stochastic models with slightly different assumptions (Distributions other than Gaussian; random walks with dependent step sizes; etc...).

We find this approach (Fig.4, Fig.6, FigS1-2) based on geometries of the phase space statistically unjustified, and unworthy to pursue in further detail. Because additional statistical details further distracts away from the biological phenomena in question. We suggest that the authors present their data in a more biological meaningful manner, and test statistical hypothesis that is succinct and straightforward. A better analysis would consist of fitting the data to a parametric model that could capture both the mean behavior of the population and variabilities among the cells (see the next point). Then within the same family of models, test for parametric differences using likelihood ratio test, or test for the necessity of a deterministic term using tools such Akaike information criterion.

See general response above.

We thank the reviewers for the suggestions. Following this advice, we have removed from the manuscript the analysis formerly depicted in Fig. 4 and Fig. 6. This was replaced by the new mathematical analysis described above, considering whether the equilibrium attractors are stable in the presence of stochasticity. We hope the reviewers will find this new perspective describes the population behavior while accounting for the presence of biologically justified stochasticity.

(iii) The presented analysis do not reject alternative models for the data. The problems we mentioned above all are related to the fact that the authors chose to ignore the stochastic aspect of their data, and treating it as statistical noise and fit everything into the box of linear models. The closeness between the data and static Gaussian model (see above) hints at the relative strength of stochastic vs dynamic forces. Other approaches such as autoregressive models or more generally Markov models could much better integrate the stochastic and dynamic aspects of the data presented. In fact, using stochastic models to understand aging and life-history traits is actively field of research, including for microbes. [20, Steiner et al 2014] Stochastic models could capture other interesting aspects of the authors' data that is very relevant for the biological points they are making. For example, the auto-correlation and cross-correlation functions of doubling times through the generations (Fig.5D) could tell us how fast do the two stable points converge with each other.

See general response above. Additionally, we have included, as requested, an autocorrelation analysis to the revision (Supplementary Fig. 4). However, we did not use the autocorrelation to the address directly the stochasticity and stability issues because we considered that the Vergassola's analysis for bounding the variance was more appropriate. Instead, we used the autocorrelation to demonstrate the leaving and entering of different bounded distributions by the doubling times of bacteria over generations.

MINOR ISSUES

(i) Abstract, line 31-32: The authors provided no evidence that the data is inconsistent with a damage-induced strategy. Molecular damages could arise stochastically in normal growing conditions.

Our intention was referring to the effect of extrinsic damage, which would induce a larger accumulation of internal damage than observed in control conditions. We apologize for the confusion and have reworded the statement as “neither a strategy induced by extrinsic damage”.

(ii) Introduction is well-written but at times misrepresents the past advances in the field:

*** Lines 43-44, 48-9: while protein damage localization to old poles was found to be correlated to fitness loss, no causal link was proven.**

After previous studies showed the correlation between damage localization and fitness loss, Winkler et al. [8], cited in both instances, concluded that there is a causal link. We have, however, removed the implication of causality from these two sentences.

*** Lines 50-51: “extrinsic damage”, referred to in ref. 11 as resulting from fluorescence light excitation was already discarded in earlier work [6] where control experiment using only phase contrast light conditions.**

In Rang et al. [11] we employed Streptomycin as a source of extrinsic damage, thus not referring solely to light excitation. Although earlier work by the reviewers [6] had successfully controlled for the potential harm produced by light excitation, Rang et al. and Coelho et al. [13] provided a relevant perspective based on entirely different sources of stress (antibiotics, heat shock, and oxidation).

*** Lines 52-53: ref. 119 does show a transient decrease of growth rate in the first few generations as old pole cell. This effect is quickly attenuated, reaching a steady state (well examined in this current manuscript!).**

Thank you. We replaced “reduction (...) over time” with “reduction (...) over hundreds of generations” to clarify.

*** Lines 54-60: “starvation could be present in the most previous bacterial aging studies” is mis-stated. Earlier works (eg, [4,6] carefully controlled for homogeneous environment of excess nutrients by: (i) quantifying equivalent growth-rate within and among concentric ‘onion shells’ of single cells and (ii) comparing growth-rate locally of sister old-new pole cells sharing the same environment. Further, in agar-pad growth, old-pole daughters are invariably pushed to the extremity of the colony rather than found at the center. Stating this, the novelty of the new experimental setup in this manuscript is a welcomed advance and indeed corroborates the results obtained with the earlier technology of past decade.**

We appreciate the careful control performed by the reviewers in previous studies. Local growth rate comparisons are a powerful tool to analyze bacteria growing as colonies. However, although starvation would hopefully affect new and old daughters alike, the environment provided by agar pads results in a greater build-up of intrinsic damage than in microfluidic devices. We have reworded our statements as “With agar pads, bacteria form mini-colonies and cells located in the center could become nutrient-limited within a few generations” and “starvation could be present in previous bacterial aging studies”.

(iii) Results

*** Fig. 2A represents the doubling time difference between the OO, ON, NN, NO populations, probed (according to the legend) by a paired t-test. The actual distributions of deltas of each**

pair (eg old - new pole actual sisters) should be reported in Supplementary to demonstrate a mean value < 0. This will assure no bias caused by outliers.

We have included the suggested analysis in Supplementary Fig. 2.

*** Fig.2D: No statistical test of spatial homogeneity is provided; too many overlapping points to make any judgements visually.**

We understand the difficulty of identifying the absence of spatial biases in Fig. 2D. This figure was meant as a graphical representation only, and the statistical test was provided in Fig. 2C through a different representation of the same data set (see Methods: Determination of cell positioning within the daughter device). We apologize for the confusion. To clarify this on the manuscript, we have inverted Fig. 2C and 2D, and emphasized their connection in the figure legend:

“(C) Representation of cells within the chamber, according to their coordinates at birth. The vertical line represents the midline of the chamber, which is open along the Y axis on both sides. (D) We analyzed the data from (C) by measuring the distance of each cell from the midline of the chamber at the moment of birth, and verified no localization bias among sibling pairs within the daughter device (OO-ON $p = 0.12$, and NO-NN $p = 0.98$, Wilcoxon Signed-Ranks Test).”

*** Fig.3A: Since the same mother cells divide to give rise a pair of old and new daughters, the effect of pole age needs to be tested comparing the pair to reflect their common heritage. As far as we could tell, this is done in Fig.2 as paired t-test, but not reflected in the linear regression models in Fig.3A. The red and blue lines seem to treat the old and new daughters as independent samples.**

The decision to treat new and old daughter doubling times as distinct subpopulations for the linear regression was determined by a two-way analysis of covariance (ANCOVA). The analysis shows that linear models for T1 and T2 are best described by distinct slopes and intercepts. To clarify this point, we moved the statistical output of this analysis from the figure legend to the main text:

“To verify whether T1 and T2 subpopulations would be better explained by individual models, we performed a two-way ANCOVA evaluating the effect of T0 and age (new or old) on daughter doubling times. While both T0 ($F = 209.15$, $p < 0.001$) and age ($F = 336.69$, $p < 0.001$) had a significant effect on T1 and T2, there was also interaction between factors ($F = 10.67$, $p = 0.001$). This indicates that the relationship between T1 or T2 and T0 is best described by distinct slopes for each subpopulation. The independent models for new ($\beta = 0.22$, $t = 8.85$, $p < 0.001$) and old daughters ($\beta = 0.35$, $t = 11.44$, $p < 0.001$) are shown in Fig. 3A.”

Compare Fig.3A-B to Fig.5B-C. The red populations in the two devices should be physiological similar, while the blue populations reflecting different stages of rejuvenation process. Does the slope being the same for the two populations in daughter devices while being different in mother machines have any biological interpretations?

Old daughter populations from the two devices are not necessarily in the same physiological stage. Because the mother machine design biases the age structure towards old lineages, the resulting population reflects longer doubling times and remarkable stability. On the other hand, old daughters produced in the

daughter device are not solely generated by stable old lineages, but also by rejuvenated mothers. This results in old daughters reflecting different stages of the aging process and exhibiting a shallower slope on the phase plane.

*** Fig.5B and Fig6A: Doubling time of the old lineage populations presented in these 2 figures seems to be much more variable than the new lineage populations, and may have long tail distributions. If true, it might suggest that the aging effect is more variable than rejuvenation, and distributed in non-symmetrical thus non-gaussian fashions. By not looking at the stochastic part of their data, the authors miss out important features of the biological effect that they are studying.**

Our previous observations indeed suggest that doubling times of old daughters exhibit slightly larger variance than new daughters. However, this is not produced by a longer tail in T2 or non-symmetrical distributions. This prediction arises naturally from the fact that old daughters inherit larger damage loads each division, resulting in a larger slope in phase plots such as Fig. 3A. Due to this relationship, rather than resulting from ignoring stochastic aspects of the data, this phenomenon would also arise from the observation of asymmetric doubling times in the complete absence of stochasticity.

(iv) Methods

*** The authors should ensure that they provided supplier information for all reagents and software used. For example, the LB components were purchased from which manufacturer? (e.g., Sigma Aldrich, St. Louis, MO, USA).**

*** details should also be provided for software including ImageJ and R.**

*** Raw data should be made available in Supplementary or deposited in a public databank. I**

This information was added to the appropriate sections.

RESPONSE TO REVIEWER #3:

I found the paper interesting and simple to follow. The experiments are cleverly designed, leading to robust results. My only suggestion is to perform few additional analysis, going beyond doubling times.

I found the discussion about the dynamics of doubling time very interesting. On the other hand, it is not clear to me what is driving that. The authors should connect that to previous literature on cell-size control. In particular, it is not obvious what is changing between old and new lineages, beyond doubling time. For instance: E. coli is known to have an adder size control. Is the added size changing between old and new lineages? Is the elongation rate changing? Do both new and old lineages have an adder behavior?

Said in other words, doubling time is an interesting quantity but is not the only worth to measure and, more importantly, duplication time alone, without information on the size and the growth rate, does not give the full story. Doubling time could be different because growth rates are different or because sizes are different or both.

We thank the reviewer for these interesting suggestions on the physiological processes driving asymmetry. We believe that we can explain at this point what drives the dynamics of doubling times. As Ariel Lindner (Reviewer #2) has shown in a published study, there is a strong relationship between the size of the aggregates of damaged proteins and the doubling time (inverse of growth rate) of old and new daughters, of different new daughters, and of different old daughters. Another study by Winkler et al. [8] has implied that damage accumulation indeed drives longer growth rates, thus representing more than just a correlation. Damage is therefore able to explain a significant part of our current differences in growth rates. Of course, eventually we want to understand the full story of what affects duplication time and other aspects of cell growth (e.g. cell size). However, we also feel that we must first establish the understanding of a reproducible starting point from which to build. As we have addressed in our response to Reviewer #1, duplication time and doubling time are highly correlated, but the variation is much higher for the former. Thus, we chose doubling times to give us the best starting point, which is directed at dynamics, stability, and stochasticity of the doubling times of bacterial lineages. As Reviewer #3 kindly indicates, our results are robust. Including other measurements at this juncture goes beyond this initial objective.

Understanding how cell size, an adder control, and the stochasticity of duplication times may be the cause(s) or a consequence(s) of our observed old and new daughter dynamics is an important and worthy endeavor that we wish to pursue as a next step. We believe that our present work establishes the needed baseline. We look forward to proceeding, and with vast and valued expertise on the adder model at UCSD (Professor Suckjoon Jun and Massimo Vergasolla), we hope to be able to complete this exciting story in the near future.

REVIEWERS' COMMENTS:

Reviewer #1 (Remarks to the Author):

The authors addressed my concerns appropriately in the revised manuscript and the reply letter. Just a minor comment:

1. Related to my previous major comment 1, the authors added new supplemental Fig. 1a and 1b, but I could not fully understand those plots due to the lack of the details. The legend says "we compared measurements acquired in separate occasions and by different collectors from mother machine cells (A and B)". Does this mean that two different individual persons analyzed the same time-lapse image sequences and compared their cell results on birth length in A and division length in B?

If that is the case, even though the results are positively correlated, cell size measurements seem to include 5 to 10% measurement errors judging from the scatter of the points from the $y=x$ lines. It would be more appropriate if the authors could confirm that those cell size measurement errors do not undermine the statistical power significantly.

Reviewer #2 (Remarks to the Author):

The authors dealt with our main concerns in an elegant and convincing manner. The new analysis is sound. Particularly convincing are the variation composition into deterministic (22% percent of the variation) vs stochastic parts (78%), the parametric formulation of the stochastic process and analysis by Prof. Vergassola (referred to in his note as discrete Ornstein–Uhlenbeck process) and the estimation of upper bounds of σ_1 and σ_2 by empirical data.

We now find that the manuscript has brought to light novel insights beyond existing literature in the field and would be of great interest to the journal's readers and the ageing basic research community at large. The new sections added to the main text are very well written and the authors took care to answer and/or position themselves convincingly on the minor points we raised. It is ripe for publication.

Yifan Yang and Ariel Lindner

RESPONSE TO REFEREES

We would like to thank the referees for their incredibly detailed feedback on our manuscript “Age structure landscapes emerge from the equilibrium between aging and rejuvenation in bacterial populations”. We believe our study has greatly improved thanks to their insights and suggestions during all stages of peer review. Our response to their last comments is detailed below:

RESPONSE TO REVIEWER #1:

The authors addressed my concerns appropriately in the revised manuscript and the reply letter. Just a minor comment:

1. Related to my previous major comment 1, the authors added new supplemental Fig. 1a and 1b, but I could not fully understand those plots due to the lack of the details. The legend says “we compared measurements acquired in separate occasions and by different collectors from mother machine cells (A and B)”. Does this mean that two different individual persons analyzed the same time-lapse image sequences and compared their cell results on birth length in A and division length in B?

This is correct. We performed a blind control where two or more individuals collected the same set of length measurements, without previous knowledge of how new and old pole inheritance could affect growth. Supplementary Fig. 1 compares some of these measurements. We have rephrased the statement to avoid ambiguity.

If that is the case, even though the results are positively correlated, cell size measurements seem to include 5 to 10% measurement errors judging from the scatter of the points from the $y=x$ lines. It would be more appropriate if the authors could confirm that those cell size measurement errors do not undermine the statistical power significantly.

The relevance of unexplained error in our measurements and statistical analysis is a valid concern, shared by the authors. Since this error results in doubling time and growth rate variation, it becomes a component of the variance that is not explained by either maternal doubling times or asymmetry, thus inflating the stochastic component. Because the fraction explained by stochasticity composes 78% of the variability in our data, measurement errors of 5% (which represents about 60% of our data) are not the main source of noise in the present study. We believe that most of the unexplained error comprised by these 78% of the variance derives from molecular stochasticity, which we intend to explore further. Unfortunately, measurement errors are inevitable for single-cell length estimates, whether they are performed by software or by hand. We are currently working in ways to mitigate these errors.

We have edited the legend of Supplementary Fig. 1 to address these concerns, namely that measurement errors become part of the variance attributed to stochasticity.

RESPONSE TO REVIEWER #2:

The authors dealt with our main concerns in an elegant and convincing manner. The new analysis is sound. Particularly convincing are the variation composition into deterministic (22% percent of the variation) vs stochastic parts (78%), the parametric formulation of the stochastic process and analysis by Prof. Vergassola (referred to in his note as discrete Ornstein–Uhlenbeck process) and the estimation of upper bounds of σ_1 and σ_2 by empirical data.

We now find that the manuscript has brought to light novel insights beyond existing literature in the field and would be of great interest to the journal's readers and the ageing basic research community at large. The new sections added to the main text are very well written and the authors took care to answer and/or position themselves convincingly on the minor points we raised. It is ripe for publication.

Yifan Yang and Ariel Lindner

We are flattered and grateful to the reviewers for their insightful comments on our study.